

# Uncovering the shortcomings of a weather typing based statistical downscaling method

Els Van Uytven[1], Jan De Niel[1], Patrick Willems[1,2]

[1]Hydraulics Section, Department of Civil Engineering, KU Leuven, Kasteelpark Arenberg 40, 3001 Leuven, Belgium
[2]Department of Hydrology and Hydraulic Engineering, Vrije Universiteit Brussel, Pleinlaan 2, 1050 Brussel, Belgium

*Correspondence to*: Els Van Uytven (els.vanuytven@kuleuven.be)

**Abstract.** In recent years many methods for statistical downscaling of the climate model outputs have been developed. Each statistical downscaling method (SDM) has strengths and limitations, but those are rarely evaluated. This paper proposes an approach to evaluate the skill of SDMs for the specific purpose of impact analysis in hydrology. The skill is evaluated by the

verification of the general statistical downscaling assumptions, and by the perfect predictor experiment that includes hydrological impact analysis. The approach has been tested for an advanced weather typing based SDM and for impact analysis on river peak flows in a Belgian river catchment. Significant shortcomings of the selected SDM were uncovered such as biases in the frequency of weather types and non-stationarities in the extreme precipitation distribution per weather type. Such evaluation of SDMs becomes of use for future tailoring of SDM ensembles to end user needs.

## 1 Introduction

For a 1.5°C temperature rise, the worldwide direct flood damage is estimated to increase in the range of 160% to 240% (Dottori et al., 2018). To minimize that potential impact, our society opts for two complementary strategies: climate mitigation and adaptation (IPCC, 2014). Consequently, vulnerability, impact and adaptation studies find ground in our society (Alfieri et al., 2016; Åström et al., 2016; Brekke et al., 2009; De Niel et al., 2018b; Termonia et al., 2018; Vansteenkiste et al., 2014a;

Vermuyten et al., 2018; Willems, 2013). These studies require projected hydro-meteorological time series, using the output of global climate models as primary information. However, the direct application of this output for impact modelling is challenged due to climate model biases (Kotlarski et al., 2014; Tabari et al., 2016) and, the mismatch in temporal and spatial resolutions between the climate model output and the time series required for impact modelling (Ochoa-Rodriguez et al., 2015; Salvadore et al., 2015). Therefore, statistical or dynamical downscaling is applied. The statistical downscaling approach bridges the

resolution gap through statistical relations between the predictors and predictand, whereas in the dynamical downscaling approach regional and limited area climate models (RCMs and LAMs) are developed. Despite the increased resolution of RCMs and LAMs, their climate model output remains biased and requires bias correction (Ehret et al., 2012; Maraun, 2016; Teutschbein and Seibert, 2012). Related to their underlying assumptions, both downscaling approaches have strengths and





limitations (Casanueva et al., 2016; Flaounas et al., 2013; Le Roux et al., 2018; Maraun et al., 2010; Vaittinada Ayar et al., 2016).

The statistical downscaling approach builds on four general assumptions (Benestad et al., 2008; Maraun and Widmann, 2018; Maraun et al., 2010; Schoof, 2013):

(1)      The informative assumption: a strong relation exists between the predictors and the predictand. It is of importance in the development of new statistical downscaling methods (SDMs), which requires the selection of predictors (Fu et al., 2018; Sachindra et al., 2018; Wilby and Wigley, 2000; Yang et al., 2017). Precipitation responds to large scale atmospheric circulation and thermo-dynamical laws (Emori and Brown, 2005; Kröner et al., 2017; Santos et al., 2016) and, hence, sea level pressure, geopotential height, relative humidity and/or (dew point) temperature are common predictors (Maraun and Widmann, 2018).

(2)      The perfect prognosis assumption: the predictors are adequately and accurately simulated by the climate models. The evaluation of this assumption is foremost performed under the name 'bias analysis'. Examples for a thorough bias analysis and the identification of underlying reasons can be found in the studies by (Anstey et al., 2013; Arakawa et al., 2014; Davini et al., 2017; Deser et al., 2012; Hartung et al., 2017; Hosseinzadehtalaei et al., 2018; Prein et al., 2015; Rybka and Tost, 2014; Tabari et al., 2016; Vanden Broucke et al., 2018; Watterson et al., 2014). These studies attribute the differences in the climate model skill to model resolution, parametrisation schemes and internal variability.

(3)      The greenhouse gas scenario sensitivity assumption: the predictors are sensitive to increasing greenhouse gas concentrations or, in other words, the predictors are greenhouse gas scenario sensitive. The greenhouse gas scenario sensitivity is measurable through trends in observed time series (Madsen et al., 2014). In climate model data, it is shown by the temporal evolution of the predictor changes (Maraun, 2013; Nguyen et al., 2018) and a comparison of the predictor changes for different representative concentration pathways (RCPs) (Jacob et al., 2014; KNMI, 2014; Van Uytven and Willems, 2018).

(4)       The stationarity assumption: the relation between the predictors and the predictand is time-invariant. Of all assumptions, this assumption is most difficult to validate (Hertig et al., 2017; Salvi et al., 2016; Wang et al., 2018). The difficulty arises from the internal variability of the climate system, which introduces anomalies in large scale circulation and precipitation (Merkenschlager et al., 2017; Ntegeka and Willems, 2008).

Alongside the main statistical downscaling assumptions, each SDM has method specific assumptions. They are encapsulated in the downscaling methodology, create the methods' strengths and limitations, and drive the statistical downscaling uncertainty contribution (Chen et al., 2011; Sunyer et al., 2015). An overview of commonly applied SDMs for precipitation downscaling and, their strengths and limitations is provided by (Hewitson et al., 2014; Maraun and Widmann, 2018; Maraun et al., 2010; Sunyer et al., 2015). In case of weather typing (WT) based SDMs, downscaling is performed in two steps. In the first step, by means of atmospheric classification systems (Philipp et al., 2016), the weather types (WTs) corresponding to the (mean sea level) pressure patterns are identified. In the second step, the relationship between the predictors (WTs) and



predictand (e.g. point precipitation) is established, using transfer functions, weather generators, re-sampling or analogues. Overall strengths emerge from the physically based WT characteristics, which represent the large scale atmospheric circulation and its changes. For analogues, limitations involve the analogues prerequisites, which divide the calibration data in smaller samples and therefore reduce the statistical reliability. Also, a historical analogue might be selected multiple times and, thus,

decrease the variance of the downscaled precipitation time series. Nevertheless, as climate projections indicate an intensification of extremes, the biggest limitation of analogues is that the produced precipitation amounts remain in the range of the historical data.

In this study, we evaluate the downscaling skill of a WT based SDM for climate change impact modelling on river peak flows

in a Belgian river catchment. The involved SDM is the WT based SDM "SD-B-7" developed by (Willems and Vrac, 2011). The downscaling skill is comprehensively evaluated through the individual verification of the general statistical downscaling assumptions and application of the perfect predictor experiment. The latter experiment has been extended by a hydrological impact analysis. In doing so, we respond to the recent focus of the statistical downscaling community on the validation of SDMs (Gutiérrez et al., 2018; Haberlandt et al., 2015; Hertig et al., 2018; Maraun et al., 2018, 2017, 2015; Pryor and Schoof,

2019; Soares et al., 2018; Werner and Cannon, 2016) and bring the validation into practice.

This study is organized as follows. Section 2 introduces the studied SDM, the selected river catchment and hydrological model applied for the rainfall-runoff simulation and, the observed and climate model simulated hydro-meteorological data. Section 3 outlines the evaluation and, corresponding results and discussions are included in Sect. 4. Section 5 summarizes the main

findings and highlights directions for future research.

## 2 Statistical downscaling method, case study and data

### 2.1 Statistical downscaling method

The SDM of study is the WT based SDM "SD-B-7" developed by (Willems and Vrac, 2011). The method downscales the daily gridded data to point time series with a sub-daily time step in three steps. In the first step, the Jenkinson-Collison

automated Lamb WT classification system is applied and the corresponding WTs are identified (Sect. 2.1.1). Next, by means of WT analogues, downscaled precipitation and potential evapotranspiration (ETo) time series are produced (Sect. 2.1.2). Finally, the precipitation amounts are scaled using the Clausius-Clapeyron relation (Sect. 2.1.3).

### 2.1.1 Jenkinson-Collison automated Lamb WT classification scheme

Based on the mean sea level pressure (MSLP) data for a 16 points grid centered around Uccle, the southerly, westerly and total

component of the geostrophic surface flow and shear velocity are calculated. Next, by inter-comparing these flow indices, 27 WTs are identified (Jenkinson and Collison, 1977; Jones et al., 1993; Philipp et al., 2016): 8 pure directional WTs (west 'W',



northwest 'NW', north 'N', northeast 'NE', east 'E', southeast 'SE', south 'S' and southwest 'SW'), 2 non-directional WTs (cyclonic WT 'C' and anti-cyclonic WT 'A'), 16 hybrid WTs (8 hybrid cyclonic-directional WTs and 8 hybrid anti-cyclonic-directional WTs) and 1 undefined WT 'U'. The corresponding MSLP patterns can be found in (Brisson et al., 2011; Demuzere et al., 2009).

The 27 WTs are regrouped to 11 WTs by equally dividing the hybrid WTs over the corresponding non-directional WTs (C or A) and directional WTs (W, NW, N, NE, E, SE, S or SW). This might lead to information loss (Schiemann and Frei, 2010); however, it is in line with previous case studies for Belgium (Brisson et al., 2011; Demuzere et al., 2009; De Niel et al., 2017; Willems and Vrac, 2011).

A detailed method description (spacing and numbering of the grid, and WT classification scheme) is included in Supplementary Information – Sect. S1.

### 2.1.2 Statistical downscaling by analogues

Downscaled time series are produced by finding analogues for the climate model output. In the first step, the bias in the number
wet days is removed using a climate model and seasonal dependent wet day threshold. In the following step, the downscaled precipitation time series are constructed by WT analogues.

In the first place, an analogue wet day is based on the season and WT. Consider day 'd' of the climate model output for the scenario period, corresponding with season 's', WT 'wt' and a daily precipitation amount 'p'. Then, the search for an analogue
day is conducted among the historical wet days in season s for which the WT equals wt. The second criterion considers the exceedance probability of the daily precipitation amount p, which is calculated using the precipitation amounts of wet days occurring in season s and corresponding with the WT wt. As such, the analogue precipitation amount for day d equals the precipitation amount of the historical day with the closest exceedance probability. The sub-daily precipitation variation corresponds to the sub-daily variation of the analogue historical day. The downscaled ETo time series are produced by the
ETo amounts corresponding to the analogue days.

### 2.1.3 Precipitation scaling by the Clausius-Clapeyron relation

Precipitation responds to large scale circulation patterns and thermo-dynamical processes. In the SDM of this study, they are accounted for by the WT analogues and precipitation scaling by the Clausius-Clapeyron relation (CC relation), respectively. The CC relation describes the water holding capacity in air masses and, more specifically, increases by 7% per degree warming.
Application of this scaling rate to precipitation intensities is valid assuming that extreme precipitation amounts are controlled by local moisture availability and are not influenced by large scale atmospheric circulation patterns. However, in reality, physical processes interact and also higher scaling rates are found (Barbero et al., 2018; Blenkinsop et al., 2018; Manola et al.,



2018; Lenderink et al., 2017; Zhang et al., 2017). The CC relation is determined on annual time scale. The temperature rise, to be applied for the CC scaling, is computed using a seasonal quantile based approach.

## 2.2 Catchment characteristics and rainfall runoff modelling

The skill of the WT based SDM is evaluated for climate change impact analysis on river peak flows (Sect. 3). For this purpose,
the Grote Nete catchment, a catchment in the northeast of Belgium is selected. The catchment covers 402 km², of which 33% is cropland, 23% forest, 19% grass land, 19% urban and built-up area. Recently, (De Niel et al., 2018b) investigated the climate change impact on the catchments' hydrological extremes. They identified a minor uncertainty contribution by the hydrological models in the peak flow changes and therefore solely NAM, a lumped conceptual hydrological model for rainfall runoff simulation, is applied in this study (DHI, 2009; Nielsen and Hansen, 1973). The NAM model has been calibrated for the Grote
Nete catchment by (Vansteenkiste et al., 2014b).

Peak flows are extracted using the Peak-Over-Threshold method of (Willems, 2009). A previous analysis (De Niel et al., 2018a) showed that, due to a high soil moisture saturation, the peak flows in the selected catchment mainly occur in the winter season. Therefore, the evaluation focusses on the winter season, which covers the months December, January and February.

## 2.3 Hydro-meteorological data

With exception to the calibration and validation of the hydrological model, for which catchment observations are employed, this study makes use of the observations for the main station of the Royal Meteorological Institute in Uccle (central Belgium; latitude = 50.8° and longitude = 4.2°). They are assumed to be representative for the selected catchment. For Uccle, precipitation, temperature and evapotranspiration (ETo) time series are available for the period 1901-2000 with a 10 minutes
and daily time step, respectively.

The historical WTs are identified using the daily gridded MSLP output for the ERA40, NCEP/NCAR and EMULATE re-analysis datasets (Table 1). Hence, this study accounts for the recent findings of (Horton and Brönnimann, 2018) and (Stryhal and Huth, 2017): both studies point to the uncertainties introduced by the re-analysis datasets in WT classification and
statistical downscaling.

The climate model ensemble, presented in Table 2, includes 93 CMIP5 climate model runs of which 33 unique control runs (Taylor et al., 2012). For the climate change impact analysis, all four representative concentration pathways (RCPs) are considered (van Vuuren et al., 2011), where the RCP 2.6, 4.5, 6.0 and 8.5 sub-ensembles include 20, 28, 15 and 30 climate
model runs, respectively. RCP 8.5 represents the business as usual scenario and corresponds with a 2.6 to 4.8°C global mean surface temperature rise (IPCC, 2014), whereas RCP 2.6 is a strong mitigation scenario for which the temperature rise by the end of the 21st century is constrained within the range 0.3-1.7°C (IPCC, 2014). For each climate model run, daily MSLP,



precipitation and average temperature output are extracted for 1961-1990 (reference/control period) and 2071-2100 (scenario period). The precipitation and temperature data are extracted for the grid cell covering Uccle, whereas MSLP, required for the WT identification, is extracted for a larger area covering Uccle (Supplementary Information – Sect. S1; Jenkinson and Collison, 1977; Jones et al., 1993; Philipp et al., 2016).

## 3 Evaluation of the statistical downscaling assumptions

### 3.1 The informative assumption

The informative assumption defines the existence of an informative relationship between the predictors and predictand. In other words, the predictors should be representative for the predictand and a physically based correlation should build this relation. In the SDM of this study, the involved predictors are the daily average temperature and WTs. By means of statistical techniques, they bridge the scale gap with the local predictand variable, this is precipitation.

The relation between WTs and precipitation across different stations in Belgium has been studied by (Brisson et al., 2011). Here we analyze the relationship between WTs and precipitation from a different point of view, i.e. the evaluation of the downscaling skill, and in this context we extend the analysis by considering multiple re-analysis datasets and focusing on the extreme precipitation amounts. In order to examine the informative assumption, the precipitation statistics related to the individual WTs are determined. The studied precipitation statistics involve the precipitation accumulation, number of wet days and empirical distribution of independent extreme precipitation amounts. The independent precipitation amounts are determined using a peak over threshold method and an inter-dependency time of minimum 12 hours (Willems, 2000). The statistics are calculated for the reference period 1961-1990 using WT and precipitation time series with a daily and 10 minutes time step, respectively. By using daily WTs, rapidly occurring changes in the large scale atmospheric circulation might be neglected (Åström et al., 2016). However, the winter season is of study and for this season no rapidly evolving circulation changes, i.e. within 1 day, are expected.

The relation between temperature and precipitation is described by the CC relation. For this study, the CC relation is verified for the independent 10 minutes independent precipitation amounts. The independent precipitation amounts are determined for the 100 years long times series using a peak over threshold method and a minimum interval between consequent events (Willems, 2000). For each event, the corresponding daily average temperature is determined. Next, following the principle described in (Manola et al., 2018), the precipitation amounts and corresponding temperatures are classified in moving temperature bins and per bin sorted from low to high. In the end, the magnification of the $90^{th}$, $95^{th}$, $99^{th}$ and $99.9^{th}$ percentile precipitation amount over increasing temperature bins is investigated.





### 3.2 The perfect prognosis assumption

The application of WT analogues follows the principle of Perfect Prognosis (PP) SDMs. For this method type, a statistical relation between observed predictors and predictand is defined and, thereafter, applied to the climate model output (Maraun et al., 2010). Consequently, the calibrated predictor-predictand relationship is only valid for a bias free simulated climate model output. The prefect prognosis assumption therefore implies an adequate and accurate simulation of both the predictors ánd the predictor-predictand relationship. More specifically, a comparison is made between the climate model simulated and observed WT occurrences and relationship between the WTs and precipitation. The other step in the SDM, the scaling of the precipitation amounts by the CC relation, is according to the principles of Model Output Statistical (MOS) SDMs. Different to PP methods, MOS methods apply predictor changes. To check whether the predictor simulation results are adequate and accurate, a comparison is made between the climate model simulated and observed daily average temperature statistics. The PP assumption moreover implies a credible predictor response to the greenhouse gas scenarios. In other words, if the climate model output would be biased, the bias should be time-invariant and not influenced by the increasing greenhouse gas concentrations. This part of the perfect prognosis assumption is not evaluated in this paper.

The climate model results considered for these evaluations are the outputs for 33 unique control runs (Table 2) and are compared against the observed results for the reference period 1961-1990. We note that the choice of the reference period (control period) and thus the internal climate variability influences the evaluation of the perfect prognosis assumption (Deser et al., 2012; Fadhel et al., 2017), in particular the WTs' associated extreme precipitation amounts.

### 3.3 The greenhouse gas scenario assumption

Referring to the greenhouse gas scenario sensitivity assumption, the predictor should respond to the greenhouse gas scenarios. Furthermore, the response may depend on the greenhouse gas scenario considered and may be magnified under increasing greenhouse gas scenarios. In this context, we analyze the change in the WT occurrences and average daily temperature in function of the four RCPs. The change is based on the model output statistics for 2071-2100 in comparison to those for 1961-1990 for an ensemble of 93 global climate models runs (Table 2). The changes for different greenhouse gas scenarios are visually inspected and focus is put on the uni-directionality and magnification of the changes for increasing greenhouse gas scenarios.

We also assess how the predictor changes contribute to the predictand changes. As such the strength of the greenhouse gas scenario sensitivity is measured, in particular for the change in the average daily precipitation amount for wet winter days. The total change accounts for the WT occurrence changes, i.e. the synoptic contribution '$\Delta P_{synoptic}$' (Eq. (1)), and remaining changes '$\Delta P_{other}$', including the thermo-dynamic and local/mesoscale feedback changes (Eq. (2)) (Souverijns et al., 2016). For both contributions, we make use of the absolute occurrence frequency of WT 'j' in the reference period $N_{j,contr}$, the absolute



occurrence frequency of WT 'j' in the scenario period $N_{j,scen}$, the average daily precipitation amount of the wet winter days associated with WT 'j' for the reference period $P_{j,contr}$ and the average daily precipitation amount of wet winter days associated with WT 'j' for the scenario period $P_{j,scen}$.

$$\Delta P_{synoptic} = \sum_{j=1}^{11}(N_{j,scen} - N_{j,contr})P_{j,contr} \ , \tag{1}$$

$$\Delta P_{other} = \sum_{j=1}^{11}(P_{j,scen} - P_{j,contr})N_{j,scen} \tag{2}$$

The evaluation of the greenhouse gas scenario sensitivity assumption is carried out assuming the bias in the climate model output is stationary and not influenced by the increasing greenhouse gas concentrations (see Sect. 3.2).

### 3.4 The stationarity assumption

The stationarity assumption requires the predictor-predictand relationship to be time-invariant or stationary. For this, we split the observed period in different sub-periods. The sub-periods are 10 years long moving windows, starting in 1901 and sliding up to 1991 by one year time step. For each of these sub-periods, the independent extreme precipitation amounts are determined and those associated with the wetter WTs are extracted. The wetter WTs were defined as part of the evaluation of the informative assumption (see Sect. 3.1). Finally, a comparison is made between the extreme precipitation amounts for different periods. This evaluation builds further on the findings of (Ntegeka and Willems, 2008), demonstrating the presence of positive and negative anomalies in the extreme precipitation amounts of the Uccle time series. Since the identification of precipitation anomalies requires long time series, the analysis is only performed for the EMULATE dataset. Once more, especially given the small length of the moving windows, we note the influence of the internal variability on precipitation extremes and its influence on the results (Deser et al., 2012; Fadhel et al., 2017).

### 3.5 The method specific assumptions

The method specific assumptions are associated with the downscaling methodology and define the SDM uncertainty. They involve the incorporation of the predictors in the statistical function and the extrapolation of the predictand values under increasing greenhouse gas scenarios For the studied SDM, the WTs and average daily temperature are included in the downscaling methodology using analogues and the CC relation, respectively. At the same time, the CC relation enables the extrapolation of the predictand variables. In the context of assessing the method specific assumptions, the SDM is applied for the perfect predictor experiment. For this experiment, the observations are divided in calibration and validation periods. The validation period is defined as 1991-2000. This period is selected over the other periods with positive precipitation anomalies because of the climate change amplification at the end of the 20[th] century (Ntegeka and Willems, 2008). This requires the application of the CC relation and therefore allows the assessment of the CC relation in practice. Different calibration periods are defined using a 10 years wide moving window, moving between 1962 and 1986 by an annual time step. The calibration period and control period are chosen equally. As such, there is no bias between the observed and climate model simulated





predictor-predictand relation. Finally, the SDM of this study is applied to the pseudo-climate model ensemble to obtain downscaled series. Ideally, the extreme precipitation amounts and peak flow discharges of the downscaled series approximate the observed ones for the period 1991-2000. However, due to the method specific assumptions, statistical downscaling adds uncertainties and deviations are expected.

## 4 Results and discussions

### 4.1 Weather type occurrences

The results on the identification of the WTs show that, generally, regardless the season and the re-analysis dataset, the A WTs occur most frequently and represent approximately 30%. For the winter season, the second highest WT occurrence is shared between the W, SW and C WTs, and are each approximately 12% (Fig. 1). Despite for some details, there are no differences in WT occurrences across the re-analysis datasets. For instance, the W occurrences for the ERA40 dataset are smaller compared to those for EMULATE and NCEP/NCAR. The difference measures approximately 3% and is at the expense of a higher occurrence of the N and NE WTs. A second difference is found for the C and U WTs, where higher occurrences are found for the EMULATE and ERA40 datasets. The WT persistency is shown in Fig. S2 in Supplementary Information – Sect. S2. Typically, frequently occurring WTs are associated with a stronger persistence. Among the frequently occurring WTs, the persistency for the A WTs varies between 10 and 15 days, whereas for the W, SW and C WTs it varies around 5 days. With exception to the extreme persistency values for the A WTs, the re-analysis datasets introduce no or limited uncertainties to the results. Regarding the A WTs, the EMULATE dataset shows deviating WT persistency for return periods higher than 3 years; higher than for the other datasets.

The WT occurrence patterns are in agreement with the recent findings of (Otero et al., 2018), apart from small differences that may be due to differences in the size and composition of the re-analysis ensemble, and the applied WT classification system (Stryhal and Huth, 2017).

### 4.2 Evaluating the informative assumption

In order to evaluate the informative assumption, the relationship between the observed WTs (predictors) and precipitation (predictand) is studied. The considered predictand statistics are the precipitation accumulation, number of wet days and independent extreme precipitation amounts. The relative winter precipitation accumulation associated per WT for the period 1961-1990 is plotted in Fig. 3. Although the A WTs occur most frequently (Fig. 1), they do not account for the highest precipitation accumulation. For the winter season, W, SW and C represent the highest share and their share to the total winter precipitation accumulation varies between 22 and 29%, 18 and 23%, and 13 and 24%, respectively. Combined, these WTs contribute on average 65% of the total winter precipitation accumulation. Different to the WT occurrences, the choice of re-analysis datasets is of importance and introduces uncertainties up to 11%. Compared to the remaining WTs, the higher





precipitation accumulation for the W, SW and C WTs is due to their higher occurrence (see Fig. 1) and higher number of wet days. As shown in Fig. 4, 89 to 93% of the W days are wet. For the SW and C days, the wet day frequency is estimated between 80 and 85% and, 73 and 87%, respectively. Additionally, on top of the higher wet day frequency, these WTs are associated with higher precipitation amounts (Fig. 5). For instance, for the NCEP/NCAR re-analysis dataset, the daily precipitation

amount with a 1 year return period associated with the W WTs measures 0.51 mm/h and is twice the size of the corresponding amount for the A WTs, which measures 0.19 mm/h. For the EMULATE dataset, a similar order of differences is found. For the ERA 40 dataset, the differences among the distributions are smaller, however still present. Also the NW WTs are characterized by higher extreme precipitation amounts and higher wet day frequency. However, compared to the W, SW and C WTs, their occurrence is rather low and therefore drives the lower precipitation accumulation.

Based on the WT occurrences (Fig. 1) and associated precipitation statistics (Fig. 3, Fig. 4 and Fig. 5), we have identified drier and wetter WTs. The W, SW and C WTs rank among the wetter WTs, while the remaining WTs are identified as the drier WTs. The distinction among the WTs is also built by MSLP composites and Belgium's geographical location. Because of the neighboring North Sea and nearby location of the Atlantic Ocean from the southwestern up to the northern border, the SW,

W, NW and N WTs transport maritime air. On the other hand, land encloses the northeastern to southern border and therefore NE, E, SE and S air flows are of continental nature and thus drier. These results are in agreement with the earlier study by (Brisson et al., 2011) investigating the spatial variation of precipitation statistics and their relation with atmospheric WTs across different locations in Belgium. Next sections mainly focus on the wetter WTs.

The relationship between precipitation and temperature is presented in Fig. 6. This figure demonstrates the intensification of the independent precipitation amounts for increasing temperatures. For instance, the 90th percentile precipitation amount increases by 7% per degree and this increase follows the CC relation. For temperatures higher than 10°C, the scaling rate increases up to 14% per degree. Similar scaling rates are obtained for the higher precipitation percentiles. We note that only extreme precipitation amounts follow these scaling rates. The relationship between precipitation and temperature found in this

study is in accordance with the earlier studies for Belgium by (De Troch, 2016) and neighbouring regions (Lenderink and Van Meijgaard, 2008).

### 4.3 Evaluating the perfect prognosis assumption

As discussed in Sect. 3.2, the perfect prognosis assumption firstly implies a bias free predictor simulation by the climate models. In order to investigate this, the inter-comparison is made between the WT occurrences for the climate model runs and

re-analysis datasets for the reference period 1961-1990 (Fig. 1). The W, SW, A and C WTs are hereby grouped among the frequently occurring WTs. However, the comparison with the re-analysis datasets reveals large biases, in particular for the W and A WTs. The climate models overestimate the occurrence of W WTs by approximately 11%, whereas the A WTs are underestimated by 14%. Moreover, different to the re-analysis datasets, the W WTs are the foremost occurring WTs in the



climate model outputs. These findings are in agreement with the recent study by (Stryhal and Huth, 2018). Using different atmospheric classification patterns, this study points to an overall overestimation of the westerly circulation, which is estimated at approximately 7% for the British isles and increases towards central Europe up to 21%. (Stryhal and Huth, 2018) also point to the poor climate model performance in reproducing the occurrence of A WTs; they found differences up to 40%. Since the

spatial scale of WTs extend the minimum skilful scale of the climate models (Benestad et al., 2008), a better agreement between the climate models and re-analysis dataset simulated WT occurrences would be expected, but internal variability plays a role as well. Although it would be possible to remove the bias in WT occurrences, the SDM in this study does not do that. Note that such bias correction would require a technique that simultaneously accounts for the bias in the WT persistence (Fig. S2 in Supplementary Information – Sect. S2) and relationship with precipitation and other hydro-meteorological variables.

The perfect prognosis assumption additionally implies a bias free simulation of the predictor-predictand relationship. In this study, the predictor-predictand relationship is firstly described by the precipitation accumulation per WT. Figure 2 shows the relative precipitation accumulation over the individual WTs for the re-analysis datasets and climate model runs for the reference period (1961-1990). The climate model runs are capable of reproducing the general features, attributing the highest

precipitation accumulation to the W, SW and C WTs. For the W WTs, the bias in the associated winter precipitation accumulation ranges between 3 and 10%, between 1 and 6% for the SW WTs % and between -2 and -13% for the C WTs. In order to further examine the bias and its drivers, the number of wet days and the empirical distribution of daily precipitation amounts for the climate model runs are compared against the results for the re-analysis datasets in Fig. 3 and Fig. 5. For the W WTs, the corresponding number of wet days and extreme precipitation amounts are underestimated. The overestimation of

winter precipitation accumulation for W WTs is, therefore, due to the large overestimation of the W WT occurrences. In contrast, the overestimation in the precipitation accumulation for the C WTs is mainly due to the overestimation of the daily precipitation amounts. Compared to the other WTs, the bias in the winter precipitation accumulation for SW WTs remains rather small. This is due the combined effect of the overestimation of SW WT occurrences and the underestimation of the number of wet days. Theoretically, the biases in the predictor-predictand relationship are also removable, but this requires a

simultaneous correction of different precipitation statistics: from the number of wet days to the precipitation amounts.

Figure 7 compares the observed and climate model simulated daily average winter temperatures in function of their empirical return period for the period 1961-1990. The climate model median is representative for the observed statistics. Hence, different to the precipitation statistics (Tabari et al., 2015), the WT occurrences and the relationship between WTs and precipitation, no

additional bias correction is required.



### 4.4 Evaluating the greenhouse gas scenario sensitivity assumption

In order to examine the greenhouse gas scenario sensitivity of the predictors, we determine the WT occurrences for 2071-2100 and compare them to 1961-1990 in function of the forcing RCP (Fig. 8 – left sub-figure). Under the total uncertainty range, i.e. all RCPs combined, the occurrence of W WTs is projected to increase by 7%, whereas for the RCP sub-ensembles the increase is magnified from 5% for RCP 2.6, to 6% for RCP 4.5, 21% for RCP 6.0 and 11% for RCP 8.5. The changes are magnified under increasing greenhouse gas scenario or RCP, except for the changes from RCP 6.0 to RCP 8.5. Also the SW WTs will occur more often, up to 11%. Also these changes are uni-directional and increase under increasing greenhouse gas scenarios, except for the changes from RCP 6.0 to RCP 8.5. These discontinuities in the uni-directionality of the changes may be explained by the smaller ensemble size for RCP 6.0 compared to the other RCPs, and/or by the different RCP sub-ensemble compositions (Table 2) and differences in the internal variability among the climate models. While the W and SW WTs will occur more often, the occurrence of the A and C WTs is projected to decrease by 11% and 3%, respectively. Also these changes are sensitive to the forcing greenhouse gas scenario. In particular, the C WTs are projected to increase by 5% for RCP 2.6, -3% for RCP 4.5, -6% for RCP 6.0 and -5% for RCP 8.5. The decrease in the A WTs is magnified from 9% for RCP 2.6, to 10% for RCP 4.5, 14% for RCP 6.0 and 12% for RCP 8.5. Overall, the frequency changes depend on the forcing RCP and, hence, the predictor (WTs and their corresponding occurrence) is sensitive to increasing greenhouse gas scenarios. Furthermore, we note that larger changes in the A occurrences typically occur with larger biases, while this is not the case for the W occurrences (Supplementary Information – Sect. S3).

Compared to the W, SW, C and A WTs, larger differences between the RCP 4.5 and RCP 8.5 sub-ensemble are found for the less frequently occurring WTs (N, NE, E, SE, S and U WTs). Because of their low occurrence rates, the relative occurrence changes are strong. The strengthened W and SW WT occurrence and decreasing C WT occurrence are in line with previous studies (Otero et al., 2018; Plavcová and Kyselý, 2013). However, the change in the A WT occurrences differs with the findings of (Otero et al., 2018). This may be due to the considered climate model ensemble, the location of the 16 points grid for the WT classification system and, the reference and scenario periods.

Also the greenhouse gas scenario sensitivity of the temperature changes is assessed. Fig. 8 (right sub-figure) presents the changes in average daily temperatures for different return periods in function of the forcing RCP. For all return periods, the temperature change increases for increasing RCP. Moreover, there are no discontinuities in the uni-directionality of the changes. For instance, the average daily temperature for a 10 years return period increases from 1.1°C for RCP 2.6, to 1.7°C for RCP 4.5, 2.4°C for RCP 6.0 and 3.5°C for RCP 8.5. Similar changes are found for the other return periods. On the other hand, the spread of the changes is return period dependent, where it obviously increases for increasing return period.





To further investigate the greenhouse gas sensitivity of the predictors and the predictor-predictand relation, the total change in the average daily winter precipitation amount is decomposed in function of its drivers (Fig. 9). The total change (median value) is estimated at 0.43mm/day, for which the synoptic changes (WT occurrence changes) contributed 20% of the total change. The majority of the total change is driven by other processes, including the thermo-dynamical processes, which explain 80%

of the total change. This is in agreement with the findings of (Kröner, 2016), indicating the importance of the thermo-dynamical processes in the winter season for Western Europe.

## 4.5 Evaluating the stationarity assumption

For the evaluation of the stationarity assumption, the predictor-predictand relationship is determined for different sub-periods in the observed period and an inter-comparison is made among these relationships. The analysis is based on the findings of

(Ntegeka and Willems, 2008), which highlighted the presence of positive and negative anomalies in the independent extreme precipitation amounts. The independent extreme precipitation amounts associated with the wetter WTs (W, SW and C individually, and all combined) have been determined for 10 years long moving windows starting between 1901 and 1991. In Fig. 10, the extreme precipitation changes are shown for the more recent moving windows and for the moving windows starting between 1931 and 1941. The latter period is related with a negative anomaly, which means that the extreme precipitation

amounts are smaller than those for moving windows starting between 1981 and 1991, which are associated with a positive anomaly (Ntegeka and Willems, 2008). Other periods among the sub-periods could be selected for the analysis (Supplementary Information – Sect. S4), but here we have chosen to highlight periods showing a strong time dependent relationship. Alike earlier findings of (Ntegeka and Willems, 2008), which were not based on a selection of WTs, the extreme precipitation amounts for moving windows starting between 1931 and 1941 are smaller compared to those for the moving windows between

1981 and 1991. The difference between the positive and negative precipitation anomaly is especially visible for the W WTs and this for all aggregation levels between 10 minutes to 1 day. More specifically, the 10 minutes precipitation amounts with a 1 year return period measures 6mm/h for the negative anomaly, whereas it is 14mm/h for the positive anomaly. Also at the daily time scale, for a 1 year return period, the precipitation amount for the positive anomaly is approximately twice the amount for the negative anomaly. For the SW WTs, the difference is only present at the sub-daily aggregation levels and return periods

larger than 1 year. For the C WTs, no differences appear between the amounts for the positive and negative anomaly. Finally, we note that uncertainties are introduced in the results for the higher return periods. They arise due to the rather short moving window and the strong effect of internal variability on the extreme precipitation amounts.

## 4.6 Evaluating the method specific assumptions

In order to evaluate the method specific assumptions, and thus the downscaling methodology, the SDM is applied for the

perfect predictor experiment. For this experiment, sub-periods in the observed period are defined as pseudo climate model runs. Next, the SDM is applied to the pseudo climate model ensemble and downscaled time series are produced. Finally, a comparison is made between the observed and downscaled series, and the downscaling methodology is evaluated based on the



reproduction of the extreme precipitation amounts (Fig. 11) and peak flow discharges (Fig. 12). We verify the presence of the common limitations of WT analogues, involving the extrapolation of predictand values and the underestimation of the predictand variance.

The median results of the climate model ensemble are close to the observed extreme precipitation amounts for all re-analysis datasets and aggregation levels (Fig. 11). Since the downscaling methodology assumes independency among consequent days and WT analogues have limitations regarding the predictand extrapolation, these results are remarkable. In particular, because of the assumed independency, an underestimation of the extreme precipitation amounts, aggregated over at least 1 day, is expected. Moreover, because of analogues, one would expect an overall underestimation of the extreme precipitation amounts.

Both types of underestimation are cancelled by precipitation scaling following the CC relation (Supplementary Information – Sect. S5), where the scaling is especially visible for the extreme 10 minutes precipitation amounts (Fig. 11). For that aggregation level, the highest observed extreme precipitation amounts are systematically lower than the downscaled ones. This might be due to randomness, which is induced by the short length of the time series. Furthermore, the CC relation removes the underestimation of the variance of the daily precipitation amounts, which is induced by the limited sampling variability among

analogue days (Supplementary Information – Sect. S5).

Different to the extreme precipitation amounts, the peak flow probability distribution is not accurately reproduced for any of the re-analysis datasets (Fig. 12). In particular, for empirical return periods smaller than 1 year, the downscaled time series underestimate the observed peak flow discharges. This underestimation reaches a maximum deviation of 2m³/s and takes place

at a return period equal to 0.4 year. An unique reason for this underestimation could not be identified. We expect contributions from the seasonal calibration of the SDM, the rather short data range for the SDM calibration and the sequencing of precipitation events and corresponding precipitation amounts. Besides precipitation, the dry periods are also of importance. They influence the soil moisture balance of the hydrological model and therefore might contribute to the peak flow underestimation. For higher return periods, the median of the pseudo climate model ensemble reproduces the observed peak

flow discharges. Furthermore, alike the reproduced precipitation amounts (Fig. 11), the spread for the pseudo climate model ensemble increases for increasing return period. The largest spread is found for the NCEP/NCAR re-analysis dataset.

Nor for the reproduced extreme precipitation amounts, neither for the peak flow discharges, systematic differences are found across the re-analysis datasets.

**5 Conclusions and outlook**

The first statistical downscaling assumption, i.e. the informative assumption, implies a strong relation between the predictor and predictand. In the SDM of this study, the predictand is precipitation and the predictors are the Jenkinson-Collison Lamb



WTs and temperature. Among all WTs, the W, C, SW and A WTs are identified as frequently occurring. More specifically, 30% of all winter days are accompanied by A WTs, whereas the other WTs occur each approximately 12%. Apart from the A WTs, these WTs are associated with a high precipitation accumulation and together explain up to 71% of the total winter precipitation amount. Respectively 91%, 83% and 79% of W, SW and C occurrences are wet days; they moreover relate to

higher precipitation extremes. The WTs are found representative and informative for the target predictand variable, precipitation. The choice of the re-analysis dataset does not influence the WT occurrence. However, when examining the predictor-predictand relationship, the re-analysis datasets show differences, for which no systematic grouping is found. Also the relationship between temperature and precipitation is found informative. More specifically, extreme precipitation amounts are magnified under increasing temperatures following the CC relation (+7%/°C). For higher temperatures, the scaling rate

evolves into the super CC relation rate (+14%/°C).

The second assumption, i.e. the perfect prognosis assumption, firstly presupposes a bias free and adequate predictor simulation. The climate model runs generally reproduce the WT occurrence pattern. However, large biases are found for the W and A occurrences. The occurrence of W WTs is overestimated by 11% at the expense of an underestimation of the A WTs. Due to

the magnitude of the bias, the W WTs become the foremost occurring WT in the climate model outputs, while in the re-analysis datasets this is the A WT. The perfect prognosis assumption also implies an adequate and accurate simulation of the predictor-predictand relation. The strong biases in the WT occurrences influence the biases in the predictor-predictand relation. As such, the overestimation of the precipitation accumulation for the W WTs is due to the overestimation of the W occurrences. Also the underestimation of the precipitation accumulation for the A WTs is driven by a bias in the occurrence frequency. This bias

is thereafter amplified by the underestimation of the wet day frequency. In contrast, for the C WTs, for which no bias in the WT occurrence nor wet day frequency is found, the bias in the associated precipitation accumulation is due to an overestimation of the daily (extreme) precipitation amounts. Since biases are identifiable, the perfect prognosis assumption is not valid, but bias correction may be possible. On the contrary, for temperature, no significant biases are identified, implying a valid perfect prognosis assumption.

The third statistical downscaling assumption concerns the predictor response to increasing RCP greenhouse gas scenarios. The WT occurrences and average daily temperature statistics are determined for 2071-2100 and are compared to the results for 1961-1990. The W WTs are projected to increase by 6% for RCP 4.5 and 11% for RCP 8.5, the SW WTs by 10% for RCP 4.5 and 14% for RCP 8.5, the A WTs by -10% for RCP 4.5 and -12% for RCP 8.5 and, the C WTs by -3% for RCP 4.5 and -5%

for RCP 8.5. The changes in the WT occurrences are generally uni-directional: they are magnified under increasing greenhouse gas scenarios. They are moreover in line with the 'wet becomes wetter' winter projections for Western Europe. Also for temperature, the change increases for increasing RCP. More specifically, for RCP 2.6 the change measures 1.3°C and increases up to 1.9°C for RCP 4.5, 2.1°C for RCP 6.0 and 3.3°C for RCP 8.5. Consequently, both predictors meet the greenhouse gas scenario sensitivity assumption. Due to greenhouse gas scenario sensitivity behavior, the changes in WT occurrences explain





20% of the change in the average precipitation amount of wet winter days, while the temperature and local/mesoscale scale changes explain 80%. These unbalanced contributions demonstrate the importance of especially temperature as predictor for precipitation downscaling.

Finally, the fourth assumption requires the predictor-predictand relationship to be time-invariant or stationary. However, we found the extreme precipitation amounts, related to the wetter WTs, to be time dependent. The end of the 20th century correlates with a positive anomaly of extreme precipitation amounts, meaning that the extreme precipitation amounts for this period are higher than the overall average of extreme precipitation amounts. Moreover, this positive anomaly is expected to be strengthened by climatic changes. For other periods, such as the period 1931-1950, a negative anomaly of extreme

precipitation amounts is detected. The stationarity assumption is known as the main caveat of the statistical downscaling approach. Therefore, the failure against the stationarity assumption is not surprising.

In addition, the method specific assumptions were evaluated in the perfect predictor experiment. The downscaled time series can accurately reproduce the extreme precipitation amounts. Hence, the application of the CC relation and the prerequisites

for an analogue day are found effective. However, for the peak flows, only the tail of the peak flow discharges, ranging from 1 year to 10 years, is accurately reproduced. For the smaller return periods, the downscaled time series underestimate the observed peak flows. This underestimation might be induced by the sequences of dry periods, which influence the moisture balance of the hydrological model and therefore the peak flows.

The studied SDM does not meet all assumptions. It is shown that the SDM has limitations and its skill should be improved. Improvements should especially address the failing perfect prognosis assumption. Since the large scale atmospheric circulation in RCMs remains biased (Addor et al., 2016; Jury et al., 2018), we suggest bias correcting WT occurrences before statistical downscaling. However, the number of WT based bias correction methods remains limited (Photiadou al., 2016). Moreover, unless taken into account, bias correction techniques may disturb the WT persistency and correlation among variables.

Uncovering the limitations of the applied WT based SDM does not mean that we discourage its' use; one should not forget that other SDMs may have other types of shortcomings and biases (Chen et al., 2011; Sunyer et al., 2015; Haberlandt et al., 2015). By considering an ensemble of SDMs, the uncertainties introduced by these limitations can be taken into account.

It is also important to mention that the above findings depend on the case study and thus the choice of the SDM and river

catchment. However, the evaluation is general and therefore applicable to other case studies. Ideally and assuming the skill is time-invariant, the SDM skill is evaluated for the current climate and thereafter accounted for in the SDM ensemble. When SDM ensembles would be considered, ensemble members could be weighted based on their skill in the perfect predictor experiment. We note the resemblance with existing climate model weighing techniques (Sanderson et al., 2017). A first step towards a weighted SDM ensemble is still to be made by the statistical downscaling community.





**Acknowledgments**

E. Van Uytven is funded by a doctoral grant from the Research Foundation – Flanders for her PhD entitled "Evaluation and improvement of statistical downscaling methods for climate change impact analysis on hydrological extremes" (F.W.O., grant number 11ZY418N). This financial support is gratefully acknowledged.

5   **Author contributions**

EVU conceptualised and developed the approach to evaluate the skill of statistical downscaling methods. Under the supervision of PW, EVU and JDN performed the formal analysis and investigation. EVU prepared the visualisation and wrote the initial draft, which was critically reviewed and revised by PW and JD.

**Data availability**

10   We acknowledge the World Climate Research Programme's Working Group on Coupled Modelling, which is responsible for CMIP, and we thank the climate modelling groups (listed in Table 2) for producing and making available their model output. The climate model data is available through the website of the Earth System Grid Federation, https://esgf.llnl.gov/. The ERA40 re-analysis data set is available through the ECMWF Public Datasets Web Interface http://apps.ecmwf.int/datasets/, the EMULATE data set through https://www.metoffice.gov.uk/hadobs/emslp/ and the NCEP/NCAR data set through

15   https://www.esrl.noaa.gov/psd/data/gridded/data.ncep.re-analayis.html. The observed precipitation time series for the Uccle station were made available by Royal Meteorological Institute.





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



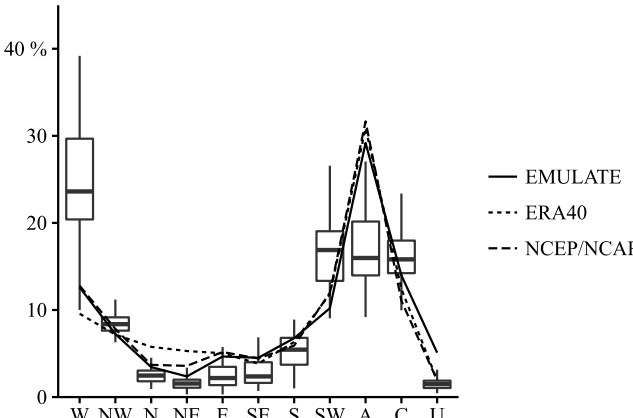

**Figure 1 Relative WT occurrence in the winter season for different re-analysis datasets (lines) and climate model runs (boxplots). The results are obtained for the reference period 1961-1990.**

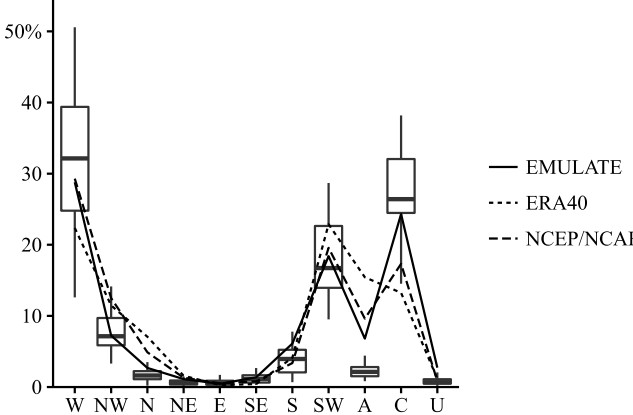

**Figure 2 Relative winter precipitation accumulation per WT for different re-analysis datasets (lines) and climate model runs (boxplots). The results are obtained for the reference period 1961-1990.**




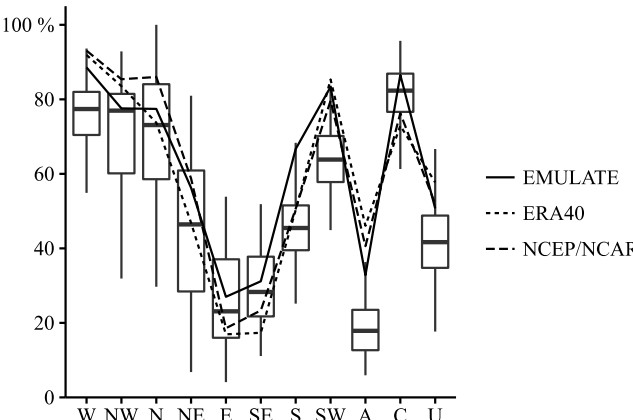

**Figure 3 Percentage wet days per WT in the winter season for different re-analysis datasets (lines) and climate model runs (boxplots). The results are obtained for the reference period 1961-1990.**

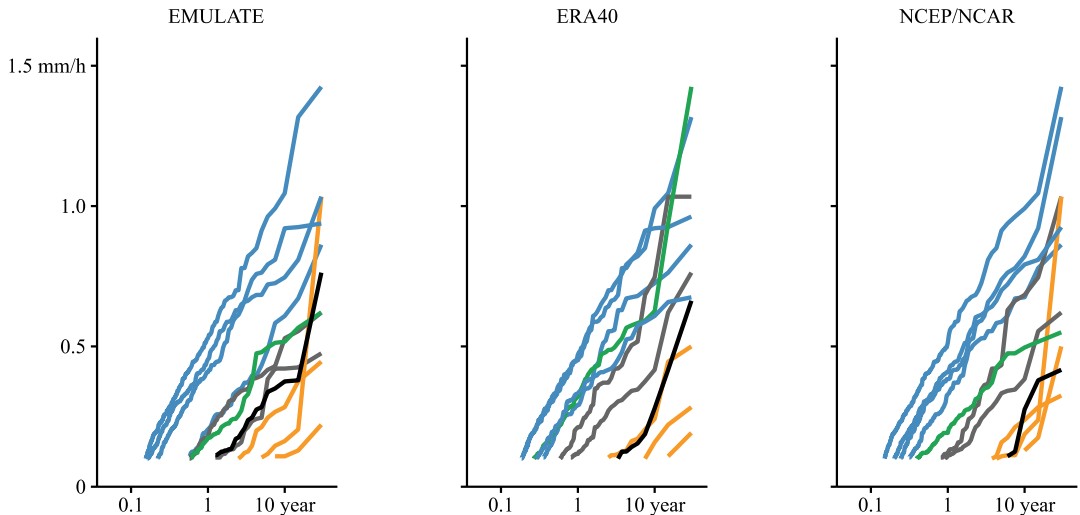

**Figure 4 Daily winter precipitation amounts per weather type for different re-analysis datasets. The blue lines indicate the results for the W, NW, SW and C WTs, the green line for the A WT, the grey lines for the N and S WTs, the orange lines for the NE, E, SE WTs and the black line for the U WTs. The results are obtained for the reference period 1961-1990.**



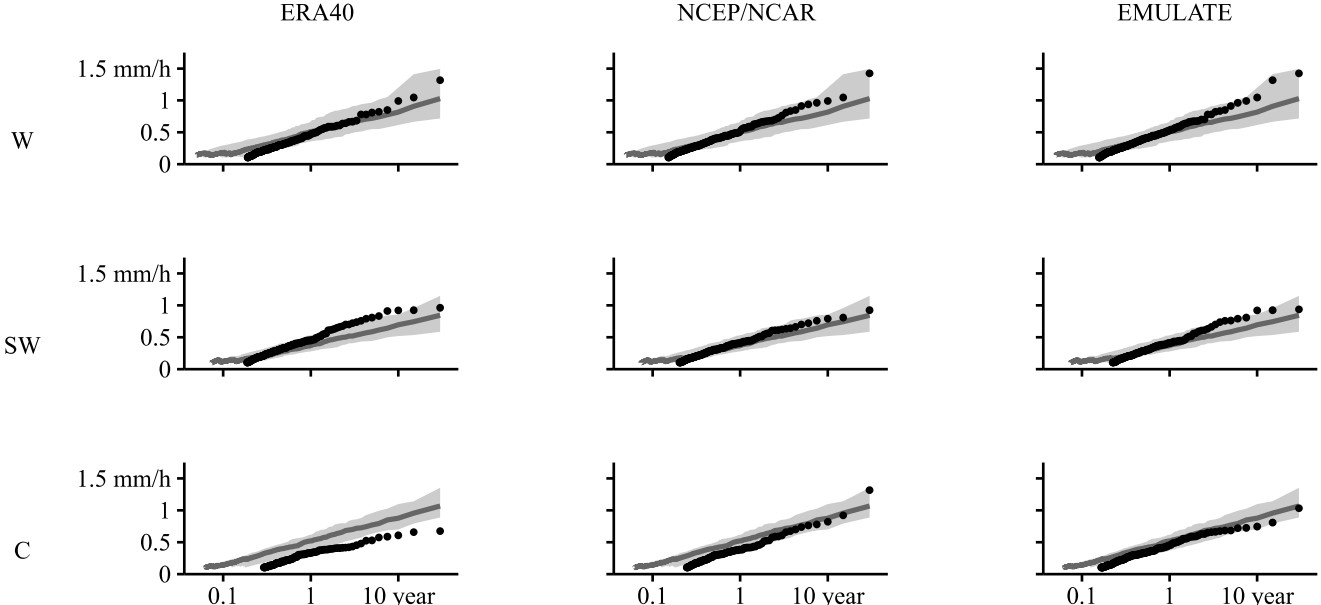

**Figure 5 Daily winter precipitation amounts in function of their empirical return period for the wettest weather types (W, SW and C WTs) and different re-analysis datasets. The black dots represent the observed relationship. The relationship simulated by the climate model runs is indicated by the grey line (median result) and area (5th – 95th percentile range). The results are obtained for the period 1961-1990.**

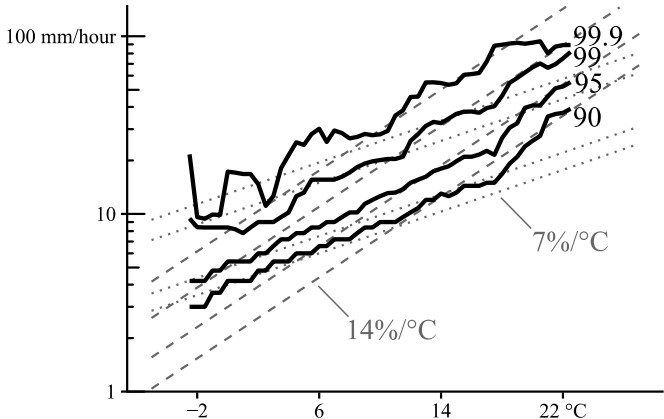

**Figure 6 The relationship between daily average temperature and independent 10 minutes precipitation amounts. The relationship is defined on annual time scale and this by using the entire Uccle time series (1901-2000). The CC relation (+7%/°C) is indicated by the grey dotted lines, whereas the 2x CC relation (+14%/°C) by the grey striped lines. The black lines show magnification of the 90th, 95th, 99th and 99.9th percentile precipitation amount for increasing temperatures.**





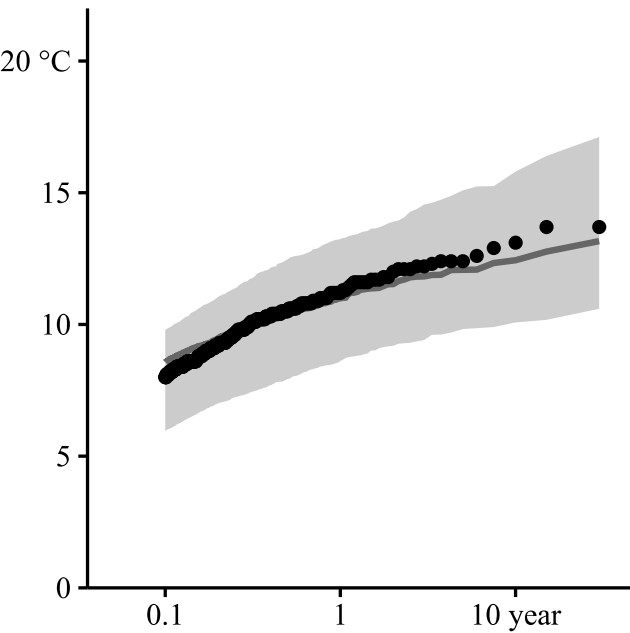

**Figure 7 Daily average temperatures in function of their empirical return period. The black dots represent the observed relationship. The relationship simulated by the climate model runs is indicated by the grey line (median result) and area (5th – 95th percentile range). The results are obtained for the reference period 1961-1990.**

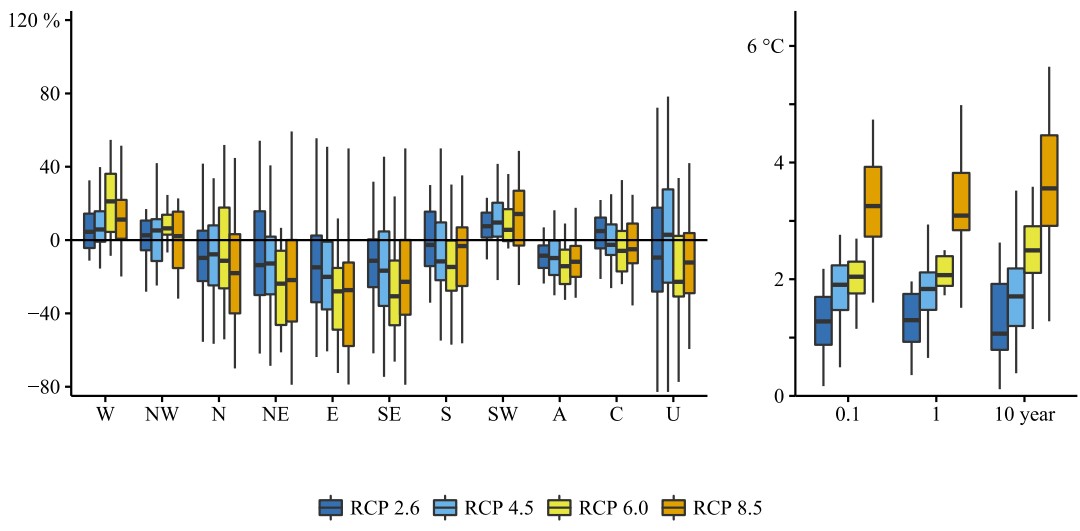

**Figure 8 Changes in the winter WT occurrences (left) and daily average temperature (right) between 2071-2100 and 1961-1990 in function of the RCP.**



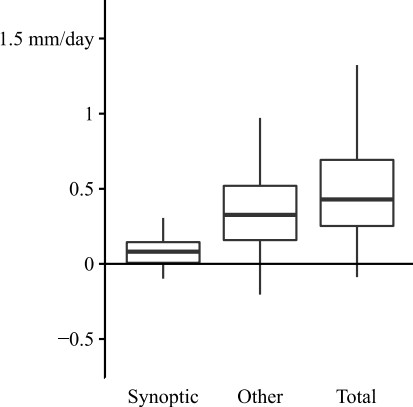

**Figure 9 Change in the average daily precipitation amount of wet winter days due to synoptic effects, i.e. changes in the WT occurrence frequencies, and other effects, for instance due to thermo-dynamical changes. The results are based on the climate model output for the scenario (2071-2100) and control period (1961-1990).**

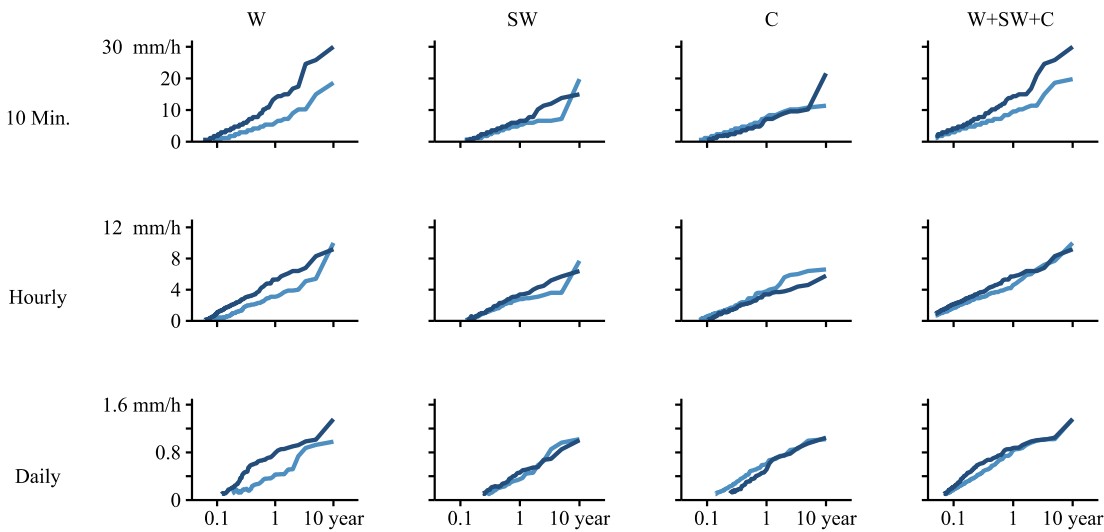

**Figure 10 Independent 10 minutes, hourly and daily aggregated winter precipitation amounts associated with the wetter WTs. The precipitation amounts for moving windows starting between 1981 and 1991 are indicated by dark blue, whereas the light blue represent the amounts for moving windows starting between 1931 and 1941. The lines show the median result.**







**Figure 11 Independent winter precipitation amounts for the observed time series (1991-2000, black dots) and downscaled time series, which are obtained in the perfect predictor experiment (grey line: median result, grey area: 5th-95th percentile range). The inter-comparison is done for different aggregation levels (rows) and re-analysis datasets (columns).**



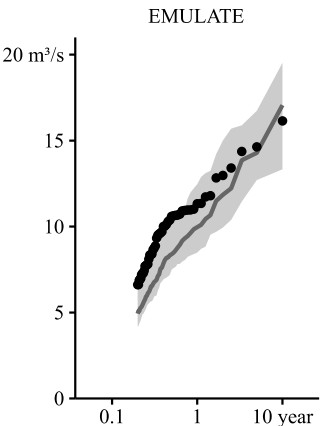
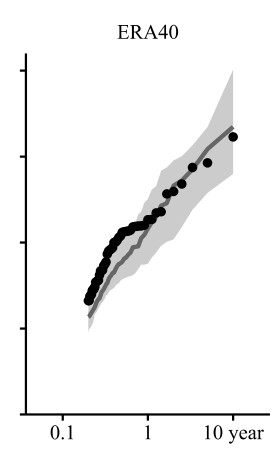
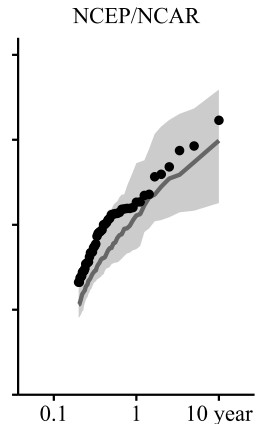

**Figure 12 Simulated, independent daily winter peak flow discharges for the observed time series (1991-2000, black dots) and downscaled time series, which are obtained in the perfect predictor experiment (grey line: median result grey area: 5th-95th percentile range). The inter-comparison is done for different re-analysis datasets (columns).**



**Table 1 Overview of the re-analysis dataset ensemble employed in this study**

| Re-analysis dataset | Resolution (Lon x Lat) | Time range | Reference |
| --- | --- | --- | --- |
| EMULATE | 5° x 5° | 1881-2000 | (Ansell et al., 2006) |
| ERA40 | 2° x 2° | 1948-present | (Uppala et al., 2005) |
| NCEP/NCAR | 2.5° x 2.5° | 1957-2002 | (Kalnay et al., 1996) |





**Table 2 Overview of the climate model ensemble employed in this study**

| Modeling Center or Group | Model name | RCP 2.6 | 4.5 | 6.0 | 8.5 | Model resolution (Lon x Lat) |
|---|---|---|---|---|---|---|
| Beijing Climate Center, China Meteorological Administration | BCC-CSM1.1(m) | | 1 | 1 | 1 | 1.1° x 1.1° |
| CSIRO (Commonwealth Scientific and Industrial Research Organisation, Australia), and BOM (Bureau of Meteorology, Australia) | ACCESS 1.0 | | | | 1 | 1.9° x 1.3° |
| Canadian Centre for Climate Modelling and Analysis | CanESM2 | 5 | 5 | | 5 | 2.8° x 2.8° |
| Centro Euro-Mediterraneo per I Cambiamenti Climatici | CMCC-CESM | | | | 1 | 3.8° x 3.7° |
| | CMCC-CMS | | 1 | | 1 | 1.9° x 1.9° |
| Centre National de Recherches Meteorologiques / Centre Europeen de Recherche et Formation Avancees en Calcul Scientifique | CNRM-CM5 | 1 | 1 | | 1 | 1.4° x 1.4° |
| Commonwealth Scientific and Industrial Research Organisation in collaboration with the Queensland Climate Change Centre of Excellence | CSIRO-Mk3.6.0 | | 1 | 1 | 1 | 1.9° x 1.9° |
| College of Global Change and Earth System Science, Beijing Normal University | BNU-ESM | 1 | 1 | | | 2.8° x 2.8° |
| Institute for Numerical Mathematics | INM-CM4 | | 1 | | 1 | 2.0° x 1.5° |
| Institut Pierre-Simon Laplace | IPSL-CM5A-LR | 1 | 3 | 1 | 3 | 3.8° x 1.9° |
| | IPSL-CM5A-MR | 1 | 1 | 1 | 1 | 2.5° x 1.3° |
| | IPSL-CM5B.LR | | 1 | | 1 | 3.8° x 1.9° |
| Japan Agency for Marine-Earth Science and Technology, Atmosphere and Ocean Research Institute (The University of Tokyo), and National Institute for Environmental Studies | MIROC-ESM-CHEM | 1 | 1 | 1 | 1 | 2.8° x 2.8° |
| | MIROC-ESM | 1 | 1 | 1 | 1 | 2.8° x 2.8° |
| Atmosphere and Ocean Research Institute (The University of Tokyo), National Institute for Environmental Studies, and Japan Agency for Marine-Earth Science and Technology | MIROC5 | 3 | 2 | 3 | 3 | 1.4° x 1.4° |
| Met Office Hadley Centre (additional HadGEM2-ES realizations contributed by Instituto Nacional de Pesquisas Espaciais) | HadGEM2-AO | 1 | 1 | 1 | 1 | 1.9° x 1.3° |
| Max Planck Institute for Meteorology (MPI-M) | MPI-ESM-LR | 1 | 1 | | 1 | 1.9° x 1.9° |
| | MPI-ESM-MR | 1 | 1 | | 1 | 1.9° x 1.9° |
| Meteorological Research Institute | MRI-CGCM3 | 1 | 1 | 1 | 1 | 1.1° x 1.1° |
| NASA Goddard Institute for Space Studies | GISS-E2-R | | 2 | | | 2.5° x 2.0° |
| Norwegian Climate Centre | NorESM1-M | 1 | | 1 | 1 | 2.5° x 1.9° |
| Geophysical Fluid Dynamics Laboratory | GFDL-CM3 | | | 1 | 1 | 2.5° x 2.0° |
| | GFDL-ESM2G | 1 | 1 | 1 | 1 | 2.5° x 2.0° |
| | GFDL-ESM2M | | 1 | 1 | 1 | 2.5° x 2.0° |