# Peer review of "Uncovering the shortcomings of a weather typing based statistical downscaling method"

_Hydrology and Earth System Sciences, 2019_

## Referee Comment (RC1) · Anonymous Referee #1 · 21 May 2019

Summary and Overall Quality: This research investigates the fidelity of a weather-typing based statistical downscaling strategy used to generate hydrometeorological forcing with respect to several of the underlying assumption implicit to these methods. In particular, they evaluate assumptions relating to the robustness of predictor-predictand relationships - their predictive power, stationarity, and sensitivity to greenhouse gas forcing - and how well those relationships are captured by coupled models. The focus of this research is a case study for downscaling of precipitation and temperature for a catchment within Belgium and makes use of an established weather-typing based downscaling strategy that also includes use of Clausius-Clapeyron (CC) scaling adjustments. The authors find informative relationships between the chosen weather-type predictors and forcing variable. While the coupled models capture the

general relationships, they exhibit significant biases in particular with respect to the frequency of the underlying weather types. The predictor-predictand relationships also exhibit non-stationarity. The authors find the use of CC-scaling adjustments result in the downscaling method being able to generate more extreme values and account for changes in variance. Overall, the manuscript is well organized, though the readability could be improved through more detailed formulation of the methods rather than the extensive narrative.

Specific Comments: 1) There is very little direct formulation of the SDM within the manuscript; it is mostly left to either supplementary material or to an extensive list of references. This left the manuscript feeling less than "self-contained," and readability could be improved with more direct formulation of the methods. This should include moving the WT-formulation from supplementary material into the primary manuscript.

2) There are a number of different datasets that are being included. However, there is very little information/discussion on why these data were selected, and it is confusing how data are being used. Why were ERA-40 and NCEP/NCAR used when these are older-generation reanalyses? The resolution of the data are disparate; how was weather typing applied to each dataset? Were they all resampled to the coarsest-resolution data (5x5) to allow for consistent WT-metrics to be defined? If not, how might the fact that the finer resolution data are likely to capture more variability affect the frequency distributions of the different WT? Were all the CMIP models resampled to the same resolution? How is the in situ, station data, being used in the compositing? Are all of the precipitation composite information being drawn using only the station data? That is, are the reanalysis only being used for developing the WT-classification and the results are just different regroupings of the underlying precipitation; or are the reanalyses precipitation actually being composited as well?

3) It is not clear if the station precipitation data can be used together with the hydrologic model. Specifically, the hydrologic model appears to have been calibrated (i.e. tuned to) a different observational dataset with likely a different climatology compared to that

of the climatology of a single station time series. This may limit the applicability of using downscaled forcing (to that of a single station) to a dataset with a different climatology than that used to calibrate the hydrologic model.

4) Results indicated super-CC scaling of precipitation changes. This indicates potentially significant components of non-thermodynamic generated forcing, either the frequency and/or intensity of weather types. The author's decomposition seems to only account for frequency changes of WT and/or precipitation changes, but is rolling-up covariant (deviation) terms into "other" forcing. A more detailed decomposition may be warranted to better understand the demonstrated super-CC scaling along with projected changes; specifically Figure 9 "other" should be more thoroughly decomposed.

5) Figure 10 is used to establish the lack of stationarity of the underlying relationships. However, the predictor-predictand relationship appears to only be evaluated with respect to temporal changes without any control for temperature changes. Given that the used SDM implements a temperature-dependent CC-scaling, it is possible that controlling for temperature changes (and CC-scaling) in addition to temporal changes may show that the utilized predictor-predictand relationship is actually stationary as long as temperature-dependency is also included. If accounting for temperature-dependent scaling related changes results in a stationary relationship, then this would provide a more robust justification for the use of CC-scaling as part of the SDM.

6) A potentially novel component of this work is related to the CC-scaling adjustments and implementation. However, it does not appear to be emphasized within the manuscript as much of the relevant material is placed in the supplementary manuscript. The scope of this work would be more novel with a stronger focus on these aspects and less on the general analysis of GCM biases in weather-type frequency and intensity; perhaps the former (CC) could be emphasized throughout the paper and the latter included in a more condensed fashion.

Technical/Grammatical Corrections:

[Figure]

Line 16: "160% to 240%" : This is confusing. Is the increase 60% to 140% of current day's values or is the increase truly 160% to 240% more than today's values (i.e. increases of 100% is a doubling of today's values). Please clearly state.

2) Line 31: "downscaling and," : There are several instances in the manuscript where the comma is placed after "and" in a compound sentence. In these cases, the "," should be placed prior to the conjunction.

2) Line 31: "by (Hewitson et al., ...)" : There are multiple instances in the manuscript where the full references are encapsulated within parentheses but should instead only have the publication year within parentheses. For example, 2)31, 3)10, and 3)23. Please carefully proofread.

6) Line 25: Redundant use of "independent"

9) Line 27: Figure 3 is noted but it should be Figure 2. Note that all figure numbers in the narrative should be double-checked.

---

## Short Comment (SC1) · 24 May 2019

This study aims at evaluating assumptions of a weather typing (WT) based statistical downscaling method (SDM) for precipitation and river peak flows in Belgium. The results of such studies provide an assessment of end user needs in choosing a right downscaling methods out of many available methods in terms of the cons and pros of each method and the intended use of the results. However, the current study for uncovering the shortcomings of a downscaling method has several serious shortcomings itself as listed below:

1. Validity of the study

The results of this study showed that the synoptic changes (WT occurrence changes)

contributed only 20% of the total change in daily precipitation and the change is mostly (80%) explained by other processes including the thermo-dynamical processes. It obviously indicates that a weather typing based statistical downscaling method shouldn't be used in the region for the downscaling of precipitation which is mainly originated from local moisture. So, what would be the point for evaluating a downscaling method which cannot be used in the region?

2. Novelty of the study

The statistical downscaling method (SDM) was taken from the literature without any modification: SD-B-7 method from Willems and Vrac (2011). The evaluation of statistical downscaling methods is not also new, although it is mentioned in the second line of the abstract that "Each statistical downscaling method (SDM) has strengths and limitations, but those are rarely evaluated". Nine more comprehensive studies were mentioned in P3L15 of this paper (some of them from EU COST Action VALUE project for the evaluation of statistical downscaling methods) along with several unmentioned. A clear objective was not defined for this study. While in the abstract, only extreme precipitation and river peak flows were stated, the majority of the paper is about daily precipitation and not extremes. As an example, it was claimed (P6L12-15) that compared to the previous study by Brisson et al. (2011) across different stations in Belgium, the current study focuses on the extreme precipitation amounts. However, the presented results are more related to winter precipitation accumulation and percentage of wet days per WT (Figures 2 and 3). Overall, there is not a consistent storyline in the paper. It appears that the paper is a combination of several small studies (leftover results actually) and then fitting a statistical downscaling to them.

3. Statistical analysis and extreme event definition

Another major issue in the paper is that the results for extremes are based on a limited sample size. The extremes were separated based on 4 seasons and 11 weather types. How extremes are the selected extremes for each season and weather type? What
threshold was used for defining extremes? Apparently, the return period of 0.1 year was chosen as the threshold. The question would be whether precipitation and streamflow that occur on average every month is really considered extreme in hydrology. Due to a small sample size after the separation per season and per weather type, even an extreme precipitation of a 10-year return period amounts 0.5 mm/hr (Figure 5).

4. Justification of the selected methods

a) Why was the Lamb weather classification used in this study, while k-means clustering is regarded as one of the best-performing classification schemes over western Europe (e.g., Beck and Philipp, 2010; Casado et al., 2010; Garcia-Valero et al., 2012; Broderick and Fealy, 2015).

b) De Niel et al. (2018b) identified a minor uncertainty contribution by the hydrological models in the peak flow changes. Among the tested hydrological models in that study, why NAM was selected for the current study?

c) Why were these three reanalysis datasets selected for this study? Why didn't they use the E-OBS observations (1950-2018) which has similar data coverage to the ERA40 reanalysis dataset (1948-2002)?

d) Several downscaling methods were developed and evaluated by Willems and Vrac (2011). Why was the SD-B-7 method selected for this study?

5. Weather types

a) The climate model results for WTs are largely biased. Although the bias was reported in the text regarding the mean scenario of the climate models, the difference goes up to 20% for the anti-cyclonic (A) WT and 30% for west (W) WT: none of the climate models can even reach the frequency of A WT estimated by the reanalysis datasets. Considering these large biases, how reliable would be the downscaling results based on these WTs? The climate models with a coarse resolution are expected to reasonably simulate these large scale patterns, and so what might be the main rea-

HESSD
son for such a bias in climate model results? I would be interesting to investigate why the GCMs have the largest uncertainty for the W WT as the main large scale driver of winter precipitation over western Europe?

b) What is the driver for the undefined weather type or the atmospheric state characterized by a weak flow? A sensitivity test of unclassified days on grid sizes and resolutions by Demuzere et al. (2009) showed that the number of these days decreases with grid resolution. Was that the case for this study as well? And generally, how will the different resolutions of reanalysis data from  $2^{\circ}$  to  $5^{\circ}$  explain the discrepancy between reanalysis-based WTs?

c) A separate set of WTs was produced for each season in the current study. What will be the influence of the seasonal cycle on the classification produced as the MSLP fields are clustered?

6. Statistical downscaling by analogues

The CMIP5 GCMs provide data at a daily time scale. Were daily precipitation data from the GCM scenario period corresponded to observed sub-daily precipitation? If so, how are the results influenced by the difference in the time scale? Were the climate change signals assumed to be time scale dependent?

7. Scaling by the Clausius-Clapeyron relation

a) Clausius-Clapeyron relation assumes that extreme precipitation amounts are controlled by local moisture availability. How local is moisture availability? The developed extreme precipitation-temperature scaling relations for central Belgium were used for river peak flow simulations in a catchment in the northeast of Belgium. How representative would be the scaling relations developed in central Belgium for northeast Belgium considering the local moisture availability assumption?

b) Dry-bulb temperature was used here for developing scaling relations, whereas several recent studies (e.g., Wasko et al., 2018) recommended to use dew point tempera-
ture than dry-bulb temperature for Clausius-Clapeyron relation, as it is a better measure of precipitation changes because of increases in the moisture holding capacity of the atmosphere (Lenderink et al., 2011).

8. Evaluation of greenhouse gas scenario assumption

To evaluate the greenhouse gas scenario assumption of the SDM, changes in the WT occurrences and average daily temperature as a function of the four RCPs were analyzed. However, this assumption might be tested for precipitation which is statistically downscaled in this study. Besides, the increase of the change in air temperature with greenhouse gas emissions is trivial. What is the relation of changes in warm extremes (with return periods ranging between 0.1 and 10 years) (Figures 7 and 8) to the downscaling of precipitation performed in this study? Though irrelevant, for warm extremes the changes in maximum temperature should be analyzed instead of mean daily temperature. P7L9-10: "To check whether the predictor simulation results are adequate and accurate, a comparison is made between the climate model simulated and observed daily average temperature statistics". Is predictor warm extremes? It seems that the full range of temperature was used for the scaling (Figure 6).

9. Evaluation of the stationarity assumption

For the evaluation of the stationarity assumption, the extreme precipitation per weather type was compared for different sub-periods of 10 years length between 1901 and 1991. Use of a 10-year sub-period for this purpose is questionable as it is far smaller than a natural climate cycle, and hence the results are greatly influenced by natural climate variability. Isn't trend analysis a more rebust approach for testing the stationarity? How large is the uncertainty in the presented results? In my view, a random selection of a dry weather type or and performing the same analysis or doing the would reveal the reliability of this results? Would performing the decadal analaysis of extreme precipitation without considereing the WTs lead to similar results? This is because based on the results of this study, weather types shouldn't be related to precipitation formation

HESSD
in the region (see comment 1).

**10. Evaluation of the method specific assumptions**

The evaluation of the SDM method specific assumptions was only performed for winter as the peak flows in the selected catchment mainly occur in the winter (P5L12-14). As mentioned in P4L30, "Application of this scaling rate to precipitation intensities is valid assuming that extreme precipitation amounts are controlled by local moisture availability and are not influenced by large scale atmospheric circulation patterns." However, the influence of the large scale circulations on winter precipitation in western Europe is well documented in the literature.

11. Interpretation and discussion of the results

a) The results were interpreted and discussed in a way that the authors expected the results be. For example, in P12L20-25, the authors attribute the difference between their results and those of Otero et al. (2018) for changes in the anti-cyclonic WT to considered climate model ensemble, the location of the 16 points grid for the WT classification system and, the reference and scenario periods. I am wondering why these differences between the two studies (this study and Otero et al., 2018) are only important for the changes in the anti-cyclonic WT and not for the changes in cyclonic, west and southwest WTs! Another example is in P15L28-31, where the authors mention only the results for RCP4.5 and RCP8.5 and not all RCPs to show that changes in the WT occurrences and precipitation are magnified under increasing greenhouse gas scenarios. Looking at the results, changes in the occurrence of W WTs are far smaller for RCP8.5 compared to RCP6.0 (P12L5). Also, there is not a clear pattern for the changes in cyclonic (C) WTs where changes are equal to 5% for RCP 2.6, - 3% for RCP 4.5, -6% for RCP 6.0 and -5% for RCP 8.5. It was speculated in P12L8-10 that these discontinuities in the uni-directionality of the changes may be explained by the smaller ensemble size for RCP 6.0 compared to the other RCPs, and/or by the different RCP sub-ensemble compositions. This issue can be easily checked by selecting
the same GCMs for different RCPs. Why isn't this an issue for temperature changes (P12L28-30).

b) In several places in the text, internal variability was argued as the reason for unexplained behaviors in the results: for example, internal variability as the reason for the large bias of climate models for WT simulations and also internal variability among the climate models as the reason for discontinuities in the uni-directionality of the changes with greenhouse gas scenarios. These statements might be supported by the results or the literature.

c) The results for the changes in the frequency of WTs showed that the frequency of the wet WTs will increase under climate change and the frequency of the dry WTs will decrease. It is worth discussing what physical explanations are behind the increasing frequency of westerlies and the decreasing frequencies of easterlies under climate change. How might global warming decrease the frequencies of cyclonic and anti-cyclonic WTs? The response to these important questions would be beneficial for improving weather typing based SDMs.

d) "Stationary is dead" is now a fact for hydrologicsts. What would be the contribution of this part of analysis to the existing knowledge? Rather than discussing the natural climate anomalies for this single location in western Europe in section 4.5, it would be more useful to discuss the drivers for such anomalies, as several global studies regarding these historical natural anomalies of the climate system have already been published. P13L21-22: "the 10 minutes precipitation amounts with a 1 year return period measures 6mm/h for the negative anomaly, whereas it is 14mm/h for the positive anomaly". What might be the reason for the positive and negative anomalies in W WT and the related extreme precipitation anomaly is especially visible for the W WTs and this for all aggregation levels between 10 minutes to 1 day. For the C WTs, no differences appear between the amounts for the positive and negative anomaly". Why is W WT-based extreme precipitation timescale-dependent, but not the C WT-based extreme

HESSD
**precipitation?**

**12. Reanalysis data uncertainty**

The uncertainty related to the choice of reanalysis data for WTs was considered. Given that the reliability of reanalysis products sharply decreases back in time (Ferguson and Villarini, 2013; Krueger et al., 2013) due to assimilating sparse observations and starting from a more uncertain initial state (Delaygue et al., 2019), a larger uncertainty of reanalysis data is expected for earlier years of the study period. Was the uncertainty calculated for the entire analysis period in this study? Does the uncertainty decrease as time progresses from far past to near past?

Specific comments P5L19: The evapotranspiration data are not daily measured data for 100 years, are they? Is it reference evapotranspiration or potential evapotranspiration? If the former is the case, with grass or alfalfa as the reference crop?

P7L15: 33 unique control runs were used in this study. I think the number of independent model runs should be lower than that!? How was the dependency between climate models investigated?

P7L16-17: The authors found that the choice of the reference period (control period) influences the evaluation of the perfect prognosis assumption. It would be interesting to present the results of the sensitivity analysis to the control period in the Supplementary Information.

Figure 12: Does grey area show the 5th-95th percentile range of climate change uncertainty?

Table 2: Were the same GCMs used for all climate variables studied?

Used references not present in the reference list of the paper

1. Beck, C., & Philipp, A. (2010). Evaluation and comparison of circulation type classifications for the European domain. Physics and Chemistry of the Earth, Parts A/B/C,
35(9-12), 374-387. 2. Broderick, C., & Fealy, R. (2015). An analysis of the synoptic and climatological applicability of circulation type classifications for Ireland. International Journal of Climatology, 35(4), 481-505. 3. Casado, M. J., Pastor, M. A., & Doblas-Reyes, F. J. (2010). Links between circulation types and precipitation over Spain. Physics and Chemistry of the Earth, Parts A/B/C, 35(9-12), 437-447. 4. Delaygue, G., Brönnimann, S., Jones, P. D., Blanchet, J., & Schwander, M. (2018). Reconstruction of Lamb weather type series back to the eighteenth century. Climate Dynamics, 1-18. 5. Ferguson, C. R., & Villarini, G. (2014). An evaluation of the statistical homogeneity of the Twentieth Century Reanalysis. Climate Dynamics, 42(11-12), 2841-2866. 6. García-Valero, J. A., Montavez, J. P., Jerez, S., Gómez-Navarro, J. J., Lorente-Plazas, R., & Jiménez-Guerrero, P. (2012). A seasonal study of the atmospheric dynamics over the Iberian Peninsula based on circulation types. Theoretical and applied climatology, 110(1-2), 291-310. 7. Krueger, O., Schenk, F., Feser, F., & Weisse, R. (2013). Inconsistencies between long-term trends in storminess derived from the 20CR reanalysis and observations. Journal of Climate, 26(3), 868-874. 8. Lenderink, G., Mok, H. Y., Lee, T. C., & Van Oldenborgh, G. J. (2011). Scaling and trends of hourly precipitation extremes in two different climate zones-Hong Kong and the Netherlands. Hydrology and Earth System Sciences, 15(9), 3033-3041. 9. Wasko, C., Lu, W. T., & Mehrotra, R. (2018). Relationship of extreme precipitation, dry-bulb temperature, and dew point temperature across Australia. Environmental Research Letters, 13(7), 074031.

---

## Referee Comment (RC2) · Anonymous Referee #2 · 26 Jul 2019

The study aims to evaluate different common assumptions inherent statistical downscaling methods (SDMs). The overarching goal of this study is certainly an important one. However more needs to be done to increase the scientific significance of this study. My few main comments are below:

(1) Why focus on WT based SDM only? Why not choose at least one more type of statistical downscaling method such as Bias-correction and Spatial downscaling method, which is a widely used SDM. I think inclusion of at least one more SDM would allow the study to more appropriately address the overarching goal.

(2) The SDM method needs to be described better. Please consider including the section S1 into the main script.

(3) Please also clarify in greater details the novelty of this study and implications beyond the study domain.

(4) I also think the manuscript can benefit from a thorough copy editing.

---

## Author Comment (AC1) · 16 Sep 2019

HESS-2019-40 Title: Uncovering shortcomings of a weather typing method statistical downscaling method Authors: Els Van Uytven, Jan De Niel and Patrick Willems

We would like to thank the reviewer for the constructive feedback and comments. Answers to the comments have been made (see below) and the manuscript has been changed accordingly (supplement to the comment, blue font). We would like to note that the manuscript has changed significantly to address the comments of all reviewers and to improve the readability.

Response to reviewer comments (Anonymous Referee #1) Summary and Overall Quality: This research investigates the fidelity of a weather typing based statistical down-

scaling strategy used to generate hydrometeorological forcing with respect to several of the underlying assumption implicit to these methods. In particular, they evaluate assumptions relating to the robustness of predictor predictand relationships - their predictive power, stationarity, and sensitivity to greenhouse gas forcing - and how well those relationships are captured by coupled models. The focus of this research is a case study for downscaling of precipitation and temperature for a catchment within Belgium and makes use of an established weather typing based downscaling strategy that also includes use of Clausius-Clapeyron (CC) scaling adjustments. The authors find informative relationships between the chosen weather-type predictors and forcing variable. While the coupled models capture the general relationships, they exhibit significant biases in particular with respect to the frequency of the underlying weather types. The predictor-predictand relationships also exhibit non-stationarity. The authors find the use of CC-scaling adjustments result in the downscaling method being able to generate more extreme values and account for changes in variance. Overall, the manuscript is well organized, though the readability could be improved through more detailed formulation of the methods rather than the extensive narrative. Specific Comments: (1) There is very little direct formulation of the SDM within the manuscript; it is mostly left to either supplementary material or to an extensive list of references. This left the manuscript feeling less than "self-contained," and readability could be improved with more direct formulation of the methods. This should include moving the WT-formulation from supplementary material into the primary manuscript.

REPLY: The WT formulation in supplementary information has moved into the main manuscript. The readability has also been improved by removing some of the redundant information in the manuscript and the more direct formulation and explanation of the methods and the results.

(2) There are a number of different datasets that are being included. However, there is very little information/discussion on why these data were selected, and it is confusing how data are being used. Why were ERA-40 and NCEP/NCAR used when these

are older-generation reanalyses? REPLY: Precipitation time series are available for the station in Uccle since 1901. We have, however, only access to the time series between 1901 and 2000. The range of this precipitation time series has been compared with the range of different re-analysis datasets. The comparison points out that the older generation re-analysis datasets cover the largest part of the available observed precipitation time series. More specifically, the re-analysis datasets and the observations have data in common for the period 1957-2000.

The resolution of the data are disparate; how was weather typing applied to each dataset? Were they all resampled to the coarsest resolution data (5x5) to allow for consistent WT-metrics to be defined? If not, how might the fact that the finer resolution data are likely to capture more variability affect the frequency distributions of the different WT? Were all the CMIP models resampled to the same resolution?

REPLY: The WTs for the re-analysis datasets and for the climate model runs have been determined considering resolution of the WT classification system. This information was originally provided in supplementary information and has been moved into the main manuscript.

How is the in situ, station data, being used in the compositing? Are all of the precipitation composite information being drawn using only the station data? That is, are the reanalysis only being used for developing the WT-classification and the results are just different regroupings of the underlying precipitation; or are the reanalyses precipitation actually being composited as well?

REPLY: The WT classification system is applied to the re-analysis datasets. Next, the produced WTs are coupled to the observed precipitation amounts, providing the historical pool with WTs and their associated precipitation amounts. We have rewritten the methodology to better explain the coupling between the precipitation time series for the RMI station in Uccle and the associated WTs based on the output of the reanalysis datasets.

(3) It is not clear if the station precipitation data can be used together with the hydrologic model. Specifically, the hydrologic model appears to have been calibrated (i.e. tuned to) a different observational dataset with likely a different climatology compared to that of the climatology of a single station time series. This may limit the applicability of using downscaled forcing (to that of a single station) to a dataset with a different climatology than that used to calibrate the hydrologic model.

REPLY: In a study for the Flemish Environment agency, De Niel and Willems (2016) investigated the spatial and temporal variations in precipitation time series for 43 rain gauges in Flanders. Their results indicated significant differences between west (coastal area) and east (Antwerp, Flemish Brabant and Limburg). As Uccle is situated in central Belgium and is located approximately 100 km from the Grote Nete catchment, the application of the hydrometeorological time series for Uccle as input series for the hydrological model of the Grote Nete catchment involves small uncertainties. The calibration of the hydrological model is, however, performed using precipitation, potential evapotranspiration and discharge time series for stations in the Grote Nete catchment.

We remark that the hydrological climate change impact analysis is removed from the manuscript for sake of brevity.

(4) Results indicated super-CC scaling of precipitation changes. This indicates potentially significant components of non-thermodynamic generated forcing, either the frequency and/or intensity of weather types. The author's decomposition seems to only account for frequency changes of WT and/or precipitation changes, but is rolling-up covariant (deviation) terms into "other" forcing. A more detailed decomposition may be warranted to better understand the demonstrated super-CC scaling along with projected changes; specifically Figure 9 "other" should be more thoroughly decomposed.

REPLY: The decomposition of the precipitation changes into contributions arising from the dynamic and thermodynamic processes has been performed for the average daily

**HESSD**

precipitation amount, projected by the climate model output. Indeed, a more detailed decomposition of the precipitation changes could be performed, as for instance done by Kröner (2016) and Kroner et al. (2017).

We would like to point out that the decomposition is performed for direct climate model output. This means that the CC relation has been indirectly considered in the climate models and is not directly applied as done by the downscaling methodology. Moreover, in the case that the results for the downscaled time series would have been used, then it is questionable whether the CC relation influences the average daily precipitation amounts. More specifically, the CC relation influences the more extreme precipitation amounts, not the average precipitation amounts.

(5) Figure 10 is used to establish the lack of stationarity of the underlying relationships. However, the predictor-predictand relationship appears to only be evaluated with respect to temporal changes without any control for temperature changes. Given that the used SDM implements a temperature-dependent CC-scaling, it is possible that controlling for temperature changes (and CC-scaling) in addition to temporal changes may show that the utilized predictor-predictand relationship is actually stationary as long as temperature-dependency is also included. If accounting for temperature-dependent scaling related changes results in a stationary relationship, then this would provide a more robust justification for the use of CC-scaling as part of the SDM. REPLY: The stationarity assumption has been evaluated using the re-analysis based WTs and the observed precipitation amounts. Hence, focus is solely put on the relation between WTs and precipitation and temperature is indeed not considered. As pointed out by the reviewer, the stationarity assumption could become more accurate when also considering temperature as a predictor. The latter could be verified by defining surrogate climate model runs and apply the SDM to the surrogate climate model runs. We however note that the empirical precipitation distributions for the W WTs for the periods 1981-1990 and 1931-1941 differ over the entire range of return periods. The application of the CC relation would thus not resolve the differences.

One of the other reviewers (Mohammad Sohrabi) wondered whether other strategies exist to test the stationarity assumption. After carefully re-reading some references, we modified the verification of the stationarity assumption. In summary, the stationarity assumption implies that the relation between the predictors and the predictand remains time-invariant. In other words, the predictors-predictand relation, which has been established using historical observations, should remain applicable under climate changes. Assuming the stationarity assumption is valid, the individual contributions by the dynamical and thermodynamic processes to the precipitation processes would not change.

In this context, the decomposition of the precipitation changes is also applied to surrogate climate model runs. The latter runs are defined by splitting the observed time series in different smaller time series. The decomposition of the surrogate climate model based precipitation amount changes is thereafter compared with the decomposition of the longterm global climate model based precipitation amount changes.

The decomposition of the surrogated based changes indicates that the contribution by the dynamical processes is important and is thus not negligible. The influence of the large scale atmospheric circulation on precipitation in winter season is comprehensively described in literature (Boé and Habets, 2014; Sousa et al., 2017; Tabari and Willems, 2018; Willems, 2013). The results furthermore indicate that the thermodymical processes gain importance at the end of the 20st century. The latter in agreement with results of Ntegeka and Willems (2008), identifying an intensification of the precipitation amounts due to the increasing temperatures.

In the modified manuscript, the approach to verify the stationarity assumption has been replaced and above discussion has been added.

(6) A potentially novel component of this work is related to the CC-scaling adjustments and implementation. However, it does not appear to be emphasized within the manuscript as much of the relevant material is placed in the supplementary manuscript.

The scope of this work would be more novel with a stronger focus on these aspects and less on the general analysis of GCM biases in weather-type frequency and intensity; perhaps the former (CC) could be emphasized throughout the paper and the latter included in a more condensed fashion.

REPLY: In the modified version of the manuscript, more focus is put on the application of the CC relation and less on the biases.   Specific Comments: (1) Line 16: "160% to 240%" : This is confusing. Is the increase 60% to 140% of current day's values or is the increase truly 160% to 240% more than today's values (i.e. increases of 100% is a doubling of today's values). Please clearly state.

REPLY: The increase is estimated between 160% and 240%. Modified.

(2) Line 31: "downscaling and," : There are several instances in the manuscript where the comma is placed after "and" in a compound sentence. In these cases, the "," should be placed prior to the conjunction.

REPLY: Modified.

(3) Line 31: "by (Hewitson et al., . . .)" : There are multiple instances in the manuscript where the full references are encapsulated within parentheses but should instead only have the publication year within parentheses. For example, 2)31, 3)10, and 3)23. Please carefully proofread.

REPLY: Modified.

(4) Line 25: Redundant use of "independent"

REPLY: Modified.

(5) Line 27: Figure 3 is noted but it should be Figure 2. Note that all figure numbers in the narrative should be double-checked.

REPLY: Modified.

References Boé, J. and Habets, F. (2014). Multi-decadal river flow variations in France. Hydrol. Earth Syst. Sci., 18, 691-708.

De Niel, J. and Willems, P. (2016). Setup of a procedure for validation and correction of meteorological data (original title: Opzetten van een validatie- en correctieprocedure voor meteorologische data), 59pp. Leuven: KU Leuven – Afdeling Hydraulica.

Kröner, N., Kotlarski, S., Fischer, E., Lüthi, D., Zubler, E. and Schär, C. (2017). Separating climate change signals into thermodynamic, lapse-rate and circulation effects: theory and application to the European summer climate. Climate Dynamics, 48 (9-10), 3425–3440.

Kröner, N. (2016). Identifying and quantifying large-scale drivers of European climate change. ETH, PhD manuscript, doi:10.3929/ethz-a-010793497.

Sousa, P. M., Trigo, R. M., Barriopedro, D., Soares, P. M. M., Ramos, A. M. and Liberato, M. L. R. (2017). Responses of European precipitation distributions and regimes to different blocking locations. Climate Dynamics, 48(3-4), 1141-1160.

Tabari, H. and Willems, P. (2018). Seasonally varying footprint of climate change on precipitation in the Middle East. Scientific Reports, 8 (4435), 1-10.

Willems, P. (2013). Multidecadal oscillatory behaviour of rainfall extremes in Europe. Climatic Change, 120 (4), 931-944.

Please also note the supplement to this comment: https://www.hydrol-earth-syst-sci-discuss.net/hess-2019-40/hess-2019-40-AC1-supplement.pdf
* * *

---

## Author Comment (AC2) · 16 Sep 2019

HESS-2019-40 Title: Uncovering shortcomings of a weather typing method statistical downscaling method Authors: Els Van Uytven, Jan De Niel and Patrick Willems

We would like to thank the reviewer for the constructive feedback and comments. Answers to the comments have been made (see below) and the manuscript has been changed accordingly (supplement to the comment, blue font). We would like to note that the manuscript has changed significantly to address the comments of all reviewers and to improve the readability.

Response to reviewer comments (Anonymous Referee #2) The study aims to evaluate different common assumptions inherent statistical downscaling methods (SDMs). The

overarching goal of this study is certainly an important one. However more needs to be done to increase the scientific significance of this study. My few main comments are below: (1) Why focus on WT based SDM only? Why not choose at least one more type of statistical downscaling method such as Bias-correction and Spatial downscaling method, which is a widely used SDM. I think inclusion of at least one more SDM would allow the study to more appropriately address the overarching goal.

REPLY: The main objective of this study is to verify and evaluate the general and structural statistical downscaling assumptions in order to develop a statistical downscaling ensemble tailored to the case study and thus end-user needs. Most studies address the general and structural statistical downscaling assumptions independently. This results in studies addressing one or some of general statistical downscaling assumptions (Dixon et al., 2016; Fu et al., 2018; Haberlandt et al., 2015; Hertig et al., 2017; Mendoza et al., 2016; Merkenschlager et al., 2017; Salvi et al., 2016; Tabari et al., 2016) and other studies addressing the structural assumptions by statistical downscaling of surrogate climate model runs (Bürger et al., 2012; Gutmann et al., 2014; Hertig et al., 2018; Maraun et al., 2018; Roberts et al., 2019; Werner and Cannon, 2016; Widmann et al., 2019; Yang et al., 2019) or by statistical downscaling of the projected climate model output (Li et al., 2017; Sørup et al., 2018; Sunyer et al., 2015; Vaittinada Ayar et al., 2016; Wang et al., 2016; Wootten et al., 2017). However, to objectively identify shortcomings of statistical downscaling methods, the verification and evaluation of the general and structural assumptions should be carried out simultaneously. To the authors knowledge, there are yet no papers which simultaneously address the verification of both types of assumptions.

We agree that verifying the general and structural assumptions for another SDM can be of interest, but such investigation is not performed in the considered study for sake of brevity.

Č (2) The SDM method needs to be described better. Please consider including the section S1 into the main script.
REPLY: The supplementary information on the methodology of the weather typing method has been placed into the main script.

(3) Please also clarify in greater details the novelty of this study and implications beyond the study domain.

REPLY: We refer the reviewer to the modified introduction and conclusions.

(4) I also think the manuscript can benefit from a thorough copy editing.

REPLY: The quality of the manuscript has been improved.

References: Bürger, G., Murdock, T. Q., Werner, A. T., Sobie, S. R., and Cannon, A. J.: Downscaling Extremes—An Intercomparison of Multiple Statistical Methods for Present Climate, Journal of Climate, 25, 4366-4388, https://doi.org/10.1175/JCLI-D-11-00408.1, 2012. Dixon, K. W., Lanzante, J. R., Nath, M. J., Hayhoe, K., Stoner, A., Radhakrishnan, A., Balaji, V., and Gaitán, C. F.: Evaluating the stationarity assumption in statistically downscaled climate projections: is past performance an indicator of future results?, Climatic Change, 135, 395-408, https://doi.org/10.1007/s10584-016-1598-0, 2016. Fu, G., Charles, S. P., Chiew, F. H., Ekström, M., and Potter, N. J.: Uncertainties of statistical downscaling from predictor selection: Equifinality and transferability, Atmospheric Research, 203, 130–140, https://doi.org/10.1016/j.atmosres.2017.12.008, 2018. Gutmann, E., Pruitt, T., Clark, M. P., Brekke, L., Arnold, J. R., Raff, D. A., and Rasmussen, R. M.: An intercomparison of statistical downscaling methods used for water resource assessments in the United States, Water Resources Research, 50, 7167-7186,https://doi.org/10.1002/2014WR015559, 2014. Haberlandt, U., Belli, A., and Bárdossy, A.: Statistical downscaling of precipitation using a stochastic rainfall model conditioned on circulation patterns - an evaluation of assumptions, International Journal of Climatology, 35, 417-432, https://doi.org/10.1002/joc.3989, 2015. Hertig, E., Merkenschlager, C., and Jacobeit, J.: Change points in predictors-predictand relationships within the scope of statistical downscaling, International Journal of ClimaInteractive comment

tology, 37, 1619–1633, https://doi.org/10.1002/joc.4801, 2017. Hertig, E., Maraun, D., Bartholy, J., Pongracz, R., Vrac, M., Mares, I., Gutiérrez, J. M., Wibig, J., Casanueva, A., and Soares, P. M. M.: Comparison of statistical downscaling methods with respect to extreme events over Europe: Validation results from the perfect predictor experiment of the COST Action VALUE, International Journal of Climatology, pp. 1-22, https://doi.org/10.1002/joc.5469, 2018. Li, J., Johnson, F., Evans, J., and Sharma, A.: A comparison of methods to estimate future sub-daily design rainfall, Advances in Water Resources, 110, 215–227, https://doi.org/10.1016/j.advwatres.2017.10.020, 2017. Maraun, D., Widmann, M., and Gutiérrez, J. M.: Statistical downscaling skill under present climate conditions: A synthesis of the VALUE perfect predictor experiment, International Journal of Climatology, pp. 1-12, https://doi.org/10.1002/joc.5877, 2018. Mendoza, P. A., Mizukami, N., Ikeda, K., Clark, M. P., Gutmann, E. D., Arnold, J. R., Brekke, L. D., and Rajagopalan, B.: Effects of different regional climate model resolution and forcing scales on projected hydrologic changes. Journal of Hydrology, 541, 1003–1019, https://doi.org/10.1016/j.jhydrol.2016.08.010, 2016. Merkenschlager, C., Hertig, E., and Jacobeit, J.: Non-stationarities in the relationships of heavy precipitation events in the Mediterranean area and the large-scale circulation in the second half of the 20th century, Global and Planetary Change, 151, 108 – 121, https://doi.org/10.1016/j.gloplacha.2016.10.009, 2017. Roberts, D. R., Wood, W. H., and Marshall, S. J.: Assessments of downscaled climate data with a high-resolution weather station network, International Journal of Climatology, 39, 3091-3103, https://doi.org/10.1002/joc.6005, 2019. Salvi, K., Ghosh, S., and Ganguly, A. R.: Credibility of statistical downscaling under nonstationary climate, Climate Dynamics, 46, 1991–2023, https://doi.org/10.1007/s00382-015-2688-9, 2016. Sørup, H. J. D., Davidsen, S., Löwe, R., Thorndahl, S. L., Borup, M., and Arnbjerg-Nielsen, K.: Evaluating catchment response to artificial rainfall from four weather generators for present and future climate, Water Science and Technology, 77, 2578-2588, https://doi.org/10.2166/wst.2018.217, 2018. Sunyer, M. A., Hundecha, Y., Lawrence, D., Madsen, H., Willems, P., Martinkova, M., Vormoor, K., Bürger, G., Hanel, M., Kri-
aučiÁnienAU, J., Loukas, A., Osuch, M., and Yücel, I.: Inter-comparison of statistical downscaling methods for projection of extreme precipitation in Europe, Hydrology and Earth System Sciences, 19, 1827–1847, https://doi.org/10.5194/hess-19-1827-2015, 2015. Tabari, H., De Troch, R., Giot, O., Hamdi, R., Termonia, P., Saeed, S., Brisson, E., Van Lipzig, N., and Willems, P.: Local impact analysis of climate change on precipitation extremes: are high-resolution climate models needed for realistic simulations?, Hvdrology and Earth System Sciences, 20, 3843-3857, https://doi.org/10.5194/hess-20-3843-2016, 2016. Vaittinada Ayar, P., Vrac, M., Bastin, S., Carreau, J., Déqué, M., and Gallardo, C.: Intercomparison of statistical and dynamical downscaling models under the EURO- and MED-CORDEX initiative framework: present climate evaluations, Climate Dynamics, 46, 1301–1329, https://doi.org/10.1007/s00382-015-2647-5, 2016. Wang, L., Ranasinghe, R., Maskey, S., van Gelder, P. H. A. J. M., and Vrijling, J. K.: Comparison of empirical statistical methods for downscaling daily climate projections from CMIP5 GCMs: A case study of the Huai River Basin, China, International Journal of Climatology, 36, 145-164, https://doi.org/10.1002/joc.4334, 2016. Werner, A. T. and Cannon, A. J.: Hydrologic extremes - an intercomparison of multiple gridded statistical downscaling methods, Hydrology and Earth System Sciences, 20, 1483–1508, https://doi.org/10.5194/hess-20-1483-2016, 2016. Widmann, M., Bedia, J., Gutiérrez, J. M., Bosshard, T., Hertig, E., Maraun, D., Casado, M. J., Ramos, P., Cardoso, R. M., Soares, P. M. M., Ribalaygua, J., Pagé, C., Fischer, A. M., Herrera, S., and Huth, R.: Validation of spatial variability in downscaling results from the VALUE perfect predictor experiment, International Journal of Climatology, pp. 1-27, https://doi.org/10.1002/joc.6024, 2019. Wootten, A., Terando, A., Reich, B. J., Boyles, R. P., and Semazzi, F.: Characterizing sources of uncertainty from global climate models and downscaling techniques, Journal of Applied Meteorology and Climatology, 56, 3245–3262, https://doi.org/10.1175/JAMC-D-17-0087.1, 2017. Yang, Y., Tang, J., Xiong, Z., Wang, S., and Yuan, J.: An intercomparison of multiple statistical downscaling methods for daily precipitation and temperature over China: present climate evaluations, Climate Dynamics, https://doi.org/10.1007/s00382-019-04809-x,
2019.

Please also note the supplement to this comment: https://www.hydrol-earth-syst-sci-discuss.net/hess-2019-40/hess-2019-40-AC2supplement.pdf

---

## Author Comment (AC3) · 16 Sep 2019

HESS-2019-40 Title: Uncovering shortcomings of a weather typing method statistical downscaling method Authors: Els Van Uytven, Jan De Niel and Patrick Willems

We would like to thank the reviewer for the constructive feedback and comments. Answers to the comments have been made (italic font) and the manuscript has been changed accordingly (blue font). We would like to note that the manuscript has changed significantly to address the comments of all reviewers and to improve the readability.

Response to reviewer comments (Mohammad Sohrabi) This study aims at evaluating assumptions of a weather typing (WT) based statistical downscaling method (SDM) for

precipitation and river peak flows in Belgium. The results of such studies provide an assessment of end user needs in choosing a right downscaling methods out of many available methods in terms of the cons and pros of each method and the intended use of the results. However, the current study for uncovering the shortcomings of a downscaling method has several serious shortcomings itself as listed below: (1) Validity of the study The results of this study showed that the synoptic changes (WT occurrence changes) contributed only 20% of the total change in daily precipitation and the change is mostly (80%) explained by other processes including the thermo-dynamical processes. It obviously indicates that a weather typing based statistical downscaling method shouldn't be used in the region for the downscaling of precipitation which is mainly originated from local moisture. So, what would be the point for evaluating a downscaling method which cannot be used in the region? REPLY: The weather typing method has been published in 2011 in an international peer reviewed and highly ranked journal (Willems and Vrac, 2011). The paper elaborates on the downscaling methodology and the differences between different weather typing methods (see also comment 4d). The paper, however, does not address the accuracy of the downscaling assumptions. Those assumptions have neither been verified in follow-up papers. On the occasion of the European VALUE COST project, the (European) statistical downscaling community has put additional focus on the evaluation of downscaling methods. In this context, the verification and evaluation of the downscaling assumptions have been conducted. Until now, no follow-up paper has indicated that the studied weather typing method has several shortcomings. These have only recently been uncovered by the verifying and evaluating the assumptions. The abstract, discussions and conclusions have been modified. They now question the applicability of the weather typing method for case studies where the precipitation changes are driven by thermodynamic processes. Indeed, Figure 9 in the original manuscript points out that the thermo-dynamical processes mainly explain the changes in the average daily precipitation amount. We note that to some extent the thermodynamic processes have been accounted for in the downscaling methodology, more specifically through the application

of the Clausius-Clapeyron relation. This relation, however, only applies to the extreme precipitation amounts, not the average daily precipitation amount. As also suggested by one of the anonymous referees, in the modified manuscript, more focus has been put to the added value of the Clausius-Clapeyron relation.

(2) Novelty of the study The statistical downscaling method (SDM) was taken from the literature without any modification: SD-B-7 method from Willems and Vrac (2011). The evaluation of statistical downscaling methods is not also new, although it is mentioned in the second line of the abstract that "Each statistical downscaling method (SDM) has strengths and limitations, but those are rarely evaluated". Nine more comprehensive studies were mentioned in P3L15 of this paper (some of them from EU COST Action VALUE project for the evaluation of statistical downscaling methods) along with several unmentioned. A clear objective was not defined for this study. While in the abstract, only extreme precipitation and river peak flows were stated, the majority of the paper is about daily precipitation and not extremes. As an example, it was claimed (P6L12-15) that compared to the previous study by Brisson et al. (2011) across different stations in Belgium, the current study focuses on the extreme precipitation amounts. However, the presented results are more related to winter precipitation accumulation and percentage of wet days per WT (Figures 2 and 3). Overall, there is not a consistent storyline in the paper. It appears that the paper is a combination of several small studies (leftover results actually) and then fitting a statistical downscaling to them. REPLY: - The introduction has been modified. The modifications, amongst others, involve a different formulation of the objective.

REPLY: The main objective of this study is to verify and evaluate the general and structural statistical downscaling assumptions in order to develop a statistical downscaling ensemble tailored to the case study and thus end-user needs. Most studies address the general and structural statistical downscaling assumptions independently. This results in studies addressing one or some of general statistical downscaling assumptions (Dixon et al., 2016; Fu et al., 2018; Haberlandt et al., 2015; Hertig et al., 2017; Mendoza et al., 2016; Merkenschlager et al., 2017; Salvi et al., 2016; Tabari et al., 2016) and other studies addressing the structural assumptions by statistical downscaling of surrogate climate model runs (Bürger et al., 2012; Gutmann et al., 2014; Hertig et al., 2018; Maraun et al., 2018; Roberts et al., 2019; Werner and Cannon, 2016; Widmann et al., 2019; Yang et al., 2019) or by statistical downscaling of the projected climate model output (Li et al., 2017; Sørup et al., 2018; Sunyer et al., 2015; Vaittinada Ayar et al., 2016; Wang et al., 2016; Wootten et al., 2017). However, to objectively identify shortcomings of statistical downscaling methods, the verification and evaluation of the general and structural assumptions should be carried out simultaneously. To the authors knowledge, there are yet no papers which simultaneously address the verification of both types of assumptions.

- The results describing the relation between the weather types and precipitation accumulation or the number of wet days have been moved to supplementary information.  (3) Statistical analysis and extreme event definition Another major issue in the paper is that the results for extremes are based on a limited sample size. The extremes were separated based on 4 seasons and 11 weather types. How extremes are the selected extremes for each season and weather type? What threshold was used for defining extremes? Apparently, the return period of 0.1 year was chosen as the threshold. The question would be whether precipitation and streamflow that occur on average every month is really considered extreme in hydrology. Due to a small sample size after the separation per season and per weather type, even an extreme precipitation of a 10-year return period amounts 0.5 mm/hr (Figure 5). REPLY: - The independent daily and sub-daily precipitation amounts are identified using a peak over threshold (POT) method. More specifically, the threshold is set at 0.1 mm/h and at least 12 hours are considered between successive events. Next the independent daily (or sub-daily precipitation amounts are classified in function of the occurrence (season) and associated weather type.

- The threshold in the POT method is not return period based, but precipitation amount

based. As indicated in above explanation, the threshold is set at 0.1 mm/h.

- The seasonal time scale has been used instead of the monthly time scale as the monthly time scale would introduce more sampling uncertainties. In order to address the sampling uncertainty, the number of WTs could be further reduced. More specifically, 4 wind directions could be considered instead of 8.

We have restricted the analysis to the winter season as De Niel et al. (2019) have shown that peak flows in the studied catchment mainly occur during winter season. We remark that the hydrological impact analysis is removed from the manuscript, but focus remains on the winter season. By considering only winter season, we focus on solely stratiform precipitation events. By combining the results for all seasons, stratiform and convective events would have been considered together. This would have been inaccurate as these events have a strongly different nature and manner of modelling. Stratiform and convective events have moreover different precipitation drivers. (4) Justification of the selected methods (a) Why was the Lamb weather classification used in this study, while k-means clustering is regarded as one of the best-performing classification schemes over western Europe (e.g., Beck and Philipp, 2010; Casado et al., 2010; Garcia-Valero et al., 2012; Broderick and Fealy, 2015). REPLY: Philipp et al. (2016) provide an overview on large scale atmospheric circulation classification systems. In summary, the classification systems are divided into subjective methods, threshold-based methods, methods based on principal component analysis, leader algorithms, hierarchical clustering analysis, optimization algorithms, mixture models and methods based on random processes. K-means methods are optimization algorithms, whereas the Jenkinson-Collison modified Lamb weather types are threshold based methods. The Jenkinson-Collison weather types have the advantage to be easier understood, as indicated in the paper of Brisson et al. (2011) and Otero et al. (2018). Brisson et al. (2011) also claim that the Jenkinson-Collison Lamb WTs are physically more correct. We would like to point out that these WTs remain presently frequently used (Ästrøm et al., 2016; Manola et al., 2019).

(b) De Niel et al. (2018) identified a minor uncertainty contribution by the hydrological models in the peak flow changes. Among the tested hydrological models in that study, why NAM was selected for the current study? REPLY: As the uncertainty contribution arising from the hydrological models is small, the reliance of the peak flow discharges does not involve large uncertainties. This is however not the case for the simulation of low flow discharges in river catchments. This has also been observed by Vansteenkiste et al. (2014), who studied the influence of hydrological model structures for the same river catchment. Moreover, the NAM rainfall runoff model is applied in many parts of the world. We would like to remark that the hydrological impact analysis has been removed from the manuscript.

(c) Why were these three reanalysis datasets selected for this study? Why didn't they use the E-OBS observations (1950-2018) which has similar data coverage to the ERA40 reanalysis dataset (1948-2002)? REPLY: E-OBS data is a land-only re-analysis dataset. For the WT typing algorithm, mean sea level pressure is also required for locations in the Atlantic Ocean and North Sea.

(d) Several downscaling methods were developed and evaluated by Willems and Vrac (2011). Why was the SD-B-7 method selected for this study? REPLY: Willems and Vrac (2011) present two types of statistical downscaling methods: precipitation change factor methods and weather typing methods. For each type, a set of methods is presented. For the weather typing methods, more specifically, differences in the methods involve the definitions for the analogue days. In total, 7 weather typing methods have been presented, of which only one method (SD-B-7) is able to produce precipitation amounts outside the range of observations. As an intensification of the precipitation extremes is expected, the extrapolation of the precipitation amounts outside the range of observations is a requirement. Consequently, Willems and Vrac (2011) advice the application of that method for climate change impact analysis.

(5) Weather types (a) The climate model results for WTs are largely biased. Although the bias was reported in the text regarding the mean scenario of the climate models,

the difference goes up to 20% for the anti-cyclonic (A) WT and 30% for west (W) WT: none of the climate models can even reach the frequency of A WT estimated by the reanalysis datasets. Considering these large biases, how reliable would be the down-scaling results based on these WTs? The climate models with a coarse resolution are expected to reasonably simulate these large scale patterns, and so what might be the main reason for such a bias in climate model results? I would be interesting to investigate why the GCMs have the largest uncertainty for the W WT as the main large scale driver of winter precipitation over western Europe? REPLY: - Climate model ensembles are often designed based on the climate model performance for current climate. However, there is yet no proof that better performing models produce more realistic projections. As stated by Mendlik and Gobiet (2016): "In the literature, models are often selected based only on their performance in the past, without regarding spread in the climate change signals, with the aim to use only the "best" models. However, correlations between past performance and future climate change signals are known to be very weak, which means that there is no clear indication that the best performing models in the past are most realistic with regard to climate change signal. In addition, the ranking of models with regard to performance in the past is highly dependent on the definition of the performance measure, which leads to a very subjective ranking." - Indeed, as discussed in the introduction of Phitan et al. (2016), the circulation biases decrease at higher horizontal climate model resolution. In order to focus on the main objective of this paper, the influence of the horizontal resolution on the WTs is not studied in the manuscript. The influence of the resolution on the biases has, however, been added to the discussion on the origin of the biases. - A brief discussion on the origin of the biases in the W and A WTs has been added to the manuscript. In summary, the North Atlantic storm track has in the climate model simulations a zonal orientation rather than SW-NE tilt (Phitan et al., 2016, Zappa et al., 2014). The zonal orientation results in a pronounced meridional pressure gradient, creating zonal westerly flows which in turn impede the occurrences of anticyclones (Stryhal and Huth, 2019). Biases in the blocking frequency might also be explained by the climate model resolution

(Anstey et al., 2013; Scaife et al., 2011, Woollings et al., 2018). (b) What is the driver for the undefined weather type or the atmospheric state characterized by a weak flow? A sensitivity test of unclassified days on grid sizes and resolutions by Demuzere et al. (2009) showed that the number of these days decreases with grid resolution. Was that the case for this study as well? And generally, how will the different resolutions of reanalysis data from 2◦ to 5◦ explain the discrepancy between reanalysis-based WTs? REPLY: - We note that the same WT classification system is applied to all seasons. The relative occurrence frequencies are, however, seasonally dependent. A difference between winter and summer would be the occurrence frequency of the undefined WTs. More specifically, the occurrence frequency of the undefined weather types in winter season is negligible, while for summer season the undefined weather types represent at least 10% of the summer days. This is in agreement with the results of Otero et al. (2018). - No sensitivity analysis on the grid resolution has been conducted as this has already performed by Demuzere et al. (2009). - The application of the 16 point grid with a 10° resolution in the zonal direction and a 5° resolution in meridional direction allows the comparison with previous studies. We note a similar motivation by Demuzere et al. (2008) and Otero et al. (2018).

(c) A separate set of WTs was produced for each season in the current study. What will be the influence of the seasonal cycle on the classification produced as the MSLP fields are clustered? REPLY: We note that the same WT classification system is applied to all seasons. The relative occurrence frequencies are, however, seasonally dependent. In order to keep the manuscript condense, only winter season is studied. We refer the reviewer also to our reply on comment 5(b).

(6) Statistical downscaling by analogues The CMIP5 GCMs provide data at a daily time scale. Were daily precipitation data from the GCM scenario period corresponded to observed sub-daily precipitation? If so, how are the results influenced by the difference in the time scale? Were the climate change signals assumed to be time scale dependent? REPLY: - In the case that the observed precipitation time series has a

sub-daily time step, the sub-daily precipitation amounts are aggregated to daily precipitation amounts. Next, for each season and WT, the exceedance probabilities for the daily precipitation amounts of wet days are calculated . In a similar way, for each of the projected wet days, the exceedance probabilities of the projected daily precipitation amounts are calculated. Thereafter, an analogue wet day is defined as an observed wet day occurring in the same season, having the same weather type and best approximating the exceedance probability of the projected day precipitation amount. To produce the downscaled time series, the daily precipitation amount for the observed analogue day is resampled. If the observed precipitation time series had a sub-daily time step, the sub-daily precipitation amounts for the analogue day are re-sampled. In other words, analogues are defined by comparing the exceedance probabilities for the observed daily precipitation amounts with the exceedance probabilities for the projected daily precipitation amounts.

- Since no sub-daily temperature time series are available, the CC relation is investigated at daily time scale. This is also the case when sub-daily precipitation amounts are available. The scaling rates identified at daily time scale are thereafter applied to the sub-daily precipitation amounts, assuming the changes at daily time scale are applicable at sub-daily time scale. (7) Scaling by the Clausius-Clapeyron relation (a) Clausius-Clapeyron relation assumes that extreme precipitation amounts are controlled by local moisture availability. How local is moisture availability? The developed extreme precipitation-temperature scaling relations for central Belgium were used for river peak flow simulations in a catchment in the northeast of Belgium. How representative would be the scaling relations developed in central Belgium for northeast Belgium considering the local moisture availability assumption?

REPLY:

In a study for the Flemish Environment agency, De Niel and Willems (2016) investigated the spatial and temporal variations in precipitation time series for 43 rain gauges in Flanders. Their results indicated significant differences between west (coastal area)

and east (Antwerp, Flemish Brabant and Limburg). A study investigating the spatial and temporal variation in the temperature time series in Flanders has yet to be conducted. As Uccle is situated in central Belgium and is located approximately 100 km from the Grote Nete catchment, it is expected that application of the hydrometeorological time series of the RMI station in Uccle for case studies in the Grote Nete catchment involves only small uncertainties. Moreover, we would like to point out that the application of the short observed time series, as is the case for the stations in the Grote Nete catchment, also involves uncertainties.

We would like to remark that the hydrological impact analysis has been removed from the manuscript. Hence, this comment is not applicable to the modified manuscript.

(b) Dry-bulb temperature was used here for developing scaling relations, whereas several recent studies (e.g., Wasko et al., 2018) recommended to use dew point temperature than dry-bulb temperature for Clausius-Clapeyron relation, as it is a better measure of precipitation changes because of increases in the moisture holding capacity of the atmosphere (Lenderink et al., 2011).

REPLY:

Indeed, several studies have pointed out that dew point temperature is a better predictor than the average daily temperature (Van de Vyver et al., 2019; Wasko et al., 2018). However, compared to average daily temperature, time series for the dew point temperature are not readily available for hydrological impact modellers.

The consideration of average daily temperatures rather than dew point temperatures has been added as a potential shortcoming of this study. (8) Evaluation of greenhouse gas scenario assumption To evaluate the greenhouse gas scenario assumption of the SDM, changes in the WT occurrences and average daily temperature as a function of the four RCPs were analyzed. However, this assumption might be tested for precipitation which is statistically downscaled in this study. Besides, the increase of the change in air temperature with greenhouse gas emissions is trivial. What is the relation of changes in warm extremes (with return periods ranging between 0.1 and 10 years) (Figures 7 and 8) to the downscaling of precipitation performed in this study? Though irrelevant, for warm extremes the changes in maximum temperature should be analyzed instead of mean daily temperature. P7L9-10: "To check whether the predictor simulation results are adequate and accurate, a comparison is made between the climate model simulated and observed daily average temperature statistics". Is predictor warm extremes? It seems that the full range of temperature was used for the scaling (Figure 6). REPLY: As suggested, the sensitivity of the predictand to the greenhouse gas scenarios and the increase in greenhouse gas scenarios is verified for the predictand. Schoof (2013) points out that some predictor variables do not respond to the greenhouse gas scenarios, while others do. This means that a predictand response is achieved by a smart choice of predictors. We note that the definition of the greenhouse gas scenario assumption has therefore been changed. The response of the WTs to the different greenhouse gas scenarios has been moved to supplementary information, but is used as background information for the discussion of the results.

(9) Evaluation of the stationarity assumption For the evaluation of the stationarity assumption, the extreme precipitation per weather type was compared for different sub-periods of 10 years length between 1901 and 1991. Use of a 10-year sub-period for this purpose is questionable as it is far smaller than a natural climate cycle, and hence the results are greatly influenced by natural climate variability. Isn't trend analysis a more robust approach for testing the stationarity? How large is the uncertainty in the presented results? In my view, a random selection of a dry weather type or and performing the same analysis or doing the would reveal the reliability of this results? Would performing the decadal analysis of extreme precipitation without considering the WTs lead to similar results? This is because based on the results of this study, weather types shouldn't be related to precipitation formation in the region (see comment 1). REPLY: - We agree that 10 years of data is rather short with respect to the natural cycle. In that context, the results are indeed influenced by the climate variability. We note that the methodology for the verification of the stationarity assumption has been

altered in the modified manuscript. In the modified manuscript, the surrogate climate model runs are 20years long. (10) Evaluation of the method specific assumptions The evaluation of the SDM method specific assumptions was only performed for winter as the peak flows in the selected catchment mainly occur in the winter (P5L12-14). As mentioned in P4L30, "Application of this scaling rate to precipitation intensities is valid assuming that extreme precipitation amounts are controlled by local moisture availability and are not influenced by large scale atmospheric circulation patterns." However, the influence of the large scale circulations on winter precipitation in western Europe is well documented in the literature. - The influence of large scale circulation patterns on the historical precipitation amounts is indeed well documented in literature (Tabari and Willems, 2018; Willems, 2013). The discussion has been extended and references have been added to the discussion of the stationarity assumption.

(11) Interpretation and discussion of the results (a) The results were interpreted and discussed in a way that the authors expected the results be. For example, in P12L20-25, the authors attribute the difference between their results and those of Otero et al. (2018) for changes in the anti-cyclonic WT to considered climate model ensemble, the location of the 16 points grid for the WT classification system and, the reference and scenario periods. I am wondering why these differences between the two studies (this study and Otero et al., 2018) are only important for the changes in the anti-cyclonic WT and not for the changes in cyclonic, west and southwest WTs! Another example is in P15L28-31, where the authors mention only the results for RCP4.5 and RCP8.5 and not all RCPs to show that changes in the WT occurrences and precipitation are magnified under increasing greenhouse gas scenarios. Looking at the results, changes in the occurrence of W WTs are far smaller for RCP8.5 compared to RCP6.0 (P12L5). Also, there is not a clear pattern for the changes in cyclonic (C) WTs where changes are equal to 5% for RCP 2.6, - 3% for RCP 4.5, -6% for RCP 6.0 and -5% for RCP 8.5. It was speculated in P12L8-10 that these discontinuities in the uni-directionality of the changes may be explained by the smaller ensemble size for RCP 6.0 compared to the other RCPs, and/or by the different RCP sub-ensemble compositions. This issue can be easily checked by selecting the same GCMs for different RCPs. Why isn't this an issue for temperature changes (P12L28-30). REPLY: - Under global warming, climate models project a poleward shift of the Northern Hemisphere jet‐streams and storm tracks, resulting increase occurrence of zonal flows and less blocking occurrences (Barnes and Screen, 2015; Santos et al., 2016; Stryhal and Huth, 2019; Woollings et al., 2018). These projections correspond with an increased occurrence of W and SW WTs and a decreased occurrence of A WTs. This physical background information has been added to the discussion of the response of the predictors and the predictand to the greenhouse gas scenarios. - Schoof (2013) points out that not all variables respond to the greenhouse gas scenarios. While the absolute temperature values increase, the absolute values of the mean sea level pressure do not change. For mean sea level pressure, changes occur in the spatial patterns. - The monotonicity of the changes for increasing greenhouse gas scenarios is not guaranteed when the ensemble compositions and sizes differ. Moreover, in some cases, for the same ensemble composition, the monotonicity of the changes for increasing greenhouse gas scenarios might be masked by the climate model uncertainties and the stochastic uncertainty, this is the uncertainty related to the variability of the climate system (Van Uytven and Willems, 2018). (b) In several places in the text, internal variability was argued as the reason for unexplained behaviors in the results: for example, internal variability as the reason for the large bias of climate models for WT simulations and also internal variability among the climate models as the reason for discontinuities in the uni-directionality of the changes with greenhouse gas scenarios. These statements might be supported by the results or the literature. - References on the influence of the influence of the internal variability and thus the choice of the reference period have been added to the manuscript. - We refer the reviewer also to our reply on comment 11a (c) The results for the changes in the frequency of WTs showed that the frequency of the wet WTs will increase under climate change and the frequency of the dry WTs will decrease. It is worth discussing what physical explanations are behind the increasing frequency of westerlies and the decreasing frequencies of easterlies under climate change. How

might global warming decrease the frequencies of cyclonic and anticyclonic WTs? The response to these important questions would be beneficial for improving weather typing based SDMs. REPLY: Under global warming, climate models project a poleward shift of the Northern Hemisphere jet‐streams and storm tracks, resulting increase occurrence of zonal flows and less blocking occurrences (Barnes and Screen, 2015; Santos et al., 2016; Stryhal and Huth, 2019; Woollings et al., 2018). Woolings et al. (2018) also list references reporting an eastward shift of the blocking activity. These physical explanations have been added to the discussion of the response of the predictand to the greenhouse gas scenarios.

(d) "Stationary is dead" is now a fact for hydrologists. What would be the contribution of this part of analysis to the existing knowledge? Rather than discussing the natural climate anomalies for this single location in western Europe in section 4.5, it would be more useful to discuss the drivers for such anomalies, as several global studies regarding these historical natural anomalies of the climate system have already been published. P13L21-22: "the 10 minutes precipitation amounts with a 1 year return period measures 6mm/h for the negative anomaly, whereas it is 14mm/h for the positive anomaly". What might be the reason for the positive and negative anomalies in W WT and the related extreme precipitation? P13L20 & P13L25: "The difference between the positive and negative precipitation anomaly is especially visible for the W WTs and this for all aggregation levels between 10 minutes to 1 day. For the C WTs, no differences appear between the amounts for the positive and negative anomaly". Why is W WT based extreme precipitation timescale-dependent, but not the C WT-based extreme precipitation? REPLY: A discussion on the drivers of the precipitation anomalies has been added to the results of the stationarity assumption. We note that the verification of the stationarity assumption has been conducted changed in the modified manuscript. (12) Reanalysis data uncertainty The uncertainty related to the choice of reanalysis data for WTs was considered. Given that the reliability of reanalysis products sharply decreases back in time (Ferguson and Villarini, 2013; Krueger et al., 2013) due to assimilating sparse observations and starting from a more uncertain initial state (Delaygue et al., 2019), a larger uncertainty of reanalysis data is expected for earlier years of the study period. Was the uncertainty calculated for the entire analysis period in this study? Does the uncertainty decrease as time progresses from far past to near past? REPLY: The re-analysis datasets provide data for different ranges of time. However, the comparison between the different re-analysis datasets is consistently performed using the same range of data. The manuscript has been modified to better indicate the prior. The time evolution of the assimilation uncertainties in the re-analysis datasets has not been studied. We agree that this can be of interest, but such investigation is not performed in the considered study for sake of brevity. We also think it would divert the scope from the current main focus.   Specific comments P5L19: The evapotranspiration data are not daily measured data for 100 years, are they? Is it reference evapotranspiration or potential evapotranspiration? If the former is the case, with grass or alfalfa as the reference crop? REPLY: We remark that the hydrological climate change impact analysis is removed from the manuscript for sake of brevity.

P7L15: 33 unique control runs were used in this study. I think the number of independent model runs should be lower than that!? How was the dependency between climate models investigated? REPLY: We agree that the number of independent model runs is smaller than the actual number of included model runs in the ensemble. The dependency between the climate model runs has not been investigated. As the word "unique" is confusing, this word has been removed from the text.

P7L16-17: The authors found that the choice of the reference period (control period) influences the evaluation of the perfect prognosis assumption. It would be interesting to present the results of the sensitivity analysis to the control period in the Supplementary Information. REPLY: While we do agree that this study can be of interest, it is not performed in the considered manuscript for sake of brevity as well as since it would divert the scope from the current main focus. References on this topic have been added to the main manuscript.

Figure 12: Does grey area show the 5th-95th percentile range of climate change uncertainty? REPLY: The 5th-95th percentile range of Figure 12 in the original manuscript represents the uncertainties associated with the surrogate climate model runs. The surrogate climate model runs are subsets of the observed time series. Hence, the 5th-95th percentile range represents the stochastic uncertainty, i.e. uncertainty related to the internal variability of the climate system.

Table 2: Were the same GCMs used for all climate variables studied? REPLY: The same GCMs were used for all climate variables studied.

Used references not present in the reference list of the paper REPLY: The reference list has been updated and completed.

References: Anstey, J. A., Davini, P., Gray, L. J., Woollings, T. J., Butchart, N., Cagnazzo, C., Christiansen, B., Hardiman, S. C., Osprey, S. M., and Yang, S.: Multi-model analysis of Northern Hemisphere winter blocking: Model biases and the role of resolution, Journal of Geophysical Research Atmospheres, 118, 3956–3971, https://doi.org/10.1002/jgrd.50231, 2013. Åström, H. L., Sunyer, M., Madsen, H., Rosbjerg, D., and Arnbjerg-Nielsen, K.: Explanatory analysis of the relationship between atmospheric circulation and occurrence of flood-generating events in a coastal city, Hydrological Processes, 30, 2773–2788, https://doi.org/10.1002/hyp.10767, 2016. Barnes, E. A. and Screen, J. A.: The impact of Arctic warming on the midlatitude jet-stream: Can it? Has it? Will it?, Wiley Interdisciplinary Reviews: Climate Change, 6, 277–286, https://doi.org/10.1002/wcc.337, 2015. Bürger, G., Murdock, T. Q., Werner, A. T., Sobie, S. R., and Cannon, A. J.: Downscaling Extremes—An Intercomparison of Multiple Statistical Methods for Present Climate, Journal of Climate, 25, 4366–4388, https://doi.org/10.1175/JCLI-D-11-00408.1, 2012. Brisson, E., Demuzere, M., Kwakernaak, B., and Van Lipzig, N. P.: Relations between atmospheric circulation and precipitation in Belgium, Meteorology and Atmospheric Physics, 111, 27–39, https://doi.org/10.1007/s00703-010-0103-y, 2011. Demuzere, M., Werner, M., van Lipzig, N. P. M., and Roeckner, E.: An analysis of present and future ECHAM5 pressure fields using a classification of circulation patterns, International Journal of Climatology,

29, 1796–1810, https://doi.org/10.1002/joc.1821, 2009. De Niel, J., Van Uytven, E., and Willems, P.: On the correlation between precipitation and potential evapotranspiration climate change signals for hydrological impact analyses. Hydrological Sciences Journal, 64 (4), 420-433. https://doi.org/10.1080/02626667.2019.1587615. 2019. De Niel, J., Van Uytven, E., and Willems, P.: Uncertainty analysis of climate change impact on river flow extremes based on a large multi model ensemble, under review, 2018. De Niel, J. and Willems, P.: Opzetten van een validatie- en correctieprocedure voor meteorologische data. Leuven: KU Leuven - Afdeling Hydraulica. 2016. . Dixon, K. W., Lanzante, J. R., Nath, M. J., Hayhoe, K., Stoner, A., Radhakrishnan, A., Balaji, V., and Gaitán, C. F.: Evaluating the stationarity assumption in statistically downscaled climate projections: is past performance an indicator of future results?, Climatic Change, 135, 395–408, https://doi.org/10.1007/s10584-016-1598-0, 2016. Fu, G., Charles, S. P., Chiew, F. H., Ekström, M., and Potter, N. J.: Uncertainties of statistical downscaling from predictor selection: Equifinality and transferability, Atmospheric Research, 203, 130–140, https://doi.org/10.1016/j.atmosres.2017.12.008, 2018. Gutmann, E., Pruitt, T., Clark, M. P., Brekke, L., Arnold, J. R., Raff, D. A., and Rasmussen, R. M.: An intercomparison of statistical downscaling methods used for water resource assessments in the United States, Water Resources Research, 50, 7167–7186,https://doi.org/10.1002/2014WR015559, 2014. Haberlandt, U., Belli, A., and Bárdossy, A.: Statistical downscaling of precipitation using a stochastic rainfall model conditioned on circulation patterns - an evaluation of assumptions, International Journal of Climatology, 35, 417–432, https://doi.org/10.1002/joc.3989, 2015. Hertig, E., Merkenschlager, C., and Jacobeit, J.: Change points in predictors– predictand relationships within the scope of statistical downscaling, International Journal of Climatology, 37, 1619–1633, https://doi.org/10.1002/joc.4801, 2017. Hertig, E., Maraun, D., Bartholy, J., Pongracz, R., Vrac, M., Mares, I., Gutiérrez, J. M., Wibig, J., Casanueva, A., and Soares, P. M. M.: Comparison of statistical downscaling methods with respect to extreme events over Europe: Validation results from the perfect predictor experiment of the COST Action VALUE, International

Journal of Climatology, pp. 1–22, https://doi.org/10.1002/joc.5469, 2018. Li, J., Johnson, F., Evans, J., and Sharma, A.: A comparison of methods to estimate future sub-daily design rainfall, Advances in Water Resources, 110, 215–227, https://doi.org/10.1016/j.advwatres.2017.10.020, 2017. Manola, I., Steeneveld, G.-J., Uijlenhoet, R. and Holtslag, A. A. M.: Analysis of urban rainfall from hourly to seasonal scales using high‐resolution radar observations in the Netherlands, 
[revised manuscript text omitted]